# Counterfactual Learning with Multioutput Deep Kernels

**Alberto Caron**                                                      *alberto.caron.19@ucl.ac.uk*
*Department of Statistical Science, University College London*
*The Alan Turing Institute, London, UK*

**Gianluca Baio**                                                              *g.baio@ucl.ac.uk*
*Department of Statistical Science, University College London*

**Ioanna Manolopoulou**                                                *i.manolopoulou@ucl.ac.uk*
*Department of Statistical Science, University College London*

**Reviewed on OpenReview:** *https: // openreview. net/ forum? id= iGREAJdULX*

## Abstract

In this paper, we address the challenge of performing counterfactual inference with ob-
servational data via Bayesian nonparametric regression adjustment, with a focus on high-
dimensional settings featuring multiple actions and multiple correlated outcomes. We present
a general class of counterfactual multi-task deep kernels models that estimate causal effects
and learn policies proficiently thanks to their sample efficiency gains, while scaling well with
high dimensions. In the first part of the work, we rely on Structural Causal Models (SCM) to
formally introduce the setup and the problem of identifying counterfactual quantities under
observed confounding. We then discuss the benefits of tackling the task of causal effects
estimation via stacked coregionalized Gaussian Processes and Deep Kernels. Finally, we
demonstrate the use of the proposed methods on simulated experiments that span individual
causal effects estimation, off-policy evaluation and optimization.

## 1 Introduction

In the recent years, there has been a surge of attention towards the use of machine learning methods for causal
inference under observational data. The desire for highly personalized decision-making is indeed pervasive in
many disciplines such as precision medicine, web advertising and education (Hodson, 2016). However, in these
fields, exploration of policies in the real-world is usually very costly and potentially harmful. For example, a
randomized clinical trial targeted at assessing the effects of a new surgical technique entails high expenses in
terms of patient recruitment and design, in addition to concerns about safety and potential side effects on
both treated and non-treated individuals. Thus, in the attempt to answer counterfactual questions about
policy interventions, such as "what would have happened if individual $i$ undertook treatment A instead of
treatment B?", one cannot typically rely on randomized experimental data. On the other hand, observational
data are abundant and more readily accessible at relatively lower costs, although they suffer from sample
selection bias issues arising because of confounding factors. Counterfactual quantities can still be identified
and estimated in settings with observed (and in some cases unobserved) confounders, provided that some
assumptions hold.

This work is aimed especially at tackling the problem of carrying out counterfactual inference using ob-
servational data, with a focus on high-dimensional settings characterized by a large number of covariates,
multiple discrete actions (or manipulative variables) and multiple correlated outcomes of interest. To this
end, we present a general class of counterfactual multi-task Deep Kernel Learning models (CounterDKL)
that efficiently adapt to multiple actions and outcomes settings by exploiting existing correlations, while
inheriting the appealing scaling properties of DKL (Wilson et al., 2016) under large samples and large number
of predictors, and preserving the Bayesian uncertainty quantification properties of Gaussian Processes (GPs).

CounterDKL essentially consists of two joint, end-to-end, components: i) a deep learning architecture that learns a lower-dimensional representation of high-dimensional input space; ii) a multitask GP (kernel-based) component (Teh et al., 2005; Bonilla et al., 2008; Álvarez et al., 2012; Bohn et al., 2019) placed on the lower-dimensional representations that can learn the posterior joint distribution of the outcomes conditional on inputs and actions, by avoiding parameter proliferation and stability issues that arise from placing a multitask GP kernel directly on the high-dimensional input space. Specifically, our contributions can be listed as follows:

- We extend the class of causal multitask GPs (Alaa & van der Schaar, 2017; 2018) to multiple actions and multiple outcomes designs, by specifying a stacked coregionalization model, highlighting its advantages in tackling "poor overlap" regions and disadvantages in terms of scalability.

- We introduce the class of CounterDKL for causal inference under multiple actions/outcomes, discussing their benefits over causal multitask GPs in high-dimensional settings (in terms of covariates, actions and outcomes).

- We demonstrate the use of CounterDKL with simulated experiments on causal effects estimation, off-policy evaluation (OPE) and learning off-policy (OPL) problems (Dudík et al., 2011; Dudík et al., 2014; Farajtabar et al., 2018; Kallus, 2021), by providing also an `Python` implementation of the models, based on `GPyTorch`[1].

Studies with multiple outcomes are quite common in applied research, as policy decisions are rarely based on a single outcome, but rather on a profile of different outcomes that might exhibit positive or negative correlation. As an example, in the medical domain, prescription of a specific treatment, such as anti-coagulants to lower cholesterol level, depends on the risk of myocardial infarction (primary outcome), as well as the risk of bleeding (unwanted, correlated, side effect).

## 1.1 Related Work

Many significant contributions in the literature on non-linear regression techniques for counterfactual/causal learning have been made in the recent years. These include works on heterogeneous (or individual) treatment effects estimation (Shalit et al., 2017; Künzel et al., 2017; Alaa & van der Schaar, 2017; 2018; Yao et al., 2018; Wager & Athey, 2018; Nie et al., 2020), and on the related field of off-policy evaluation and learning (Dudík et al., 2014; Farajtabar et al., 2018; Kallus, 2018; Athey & Wager, 2021), that focus more on the prescriptive, rather than predictive, goal. Among the several contributions, we particularly draw attention to the early work of Hill (2011) that first proposed Bayesian nonparametric methods (specifically Bayesian Additive Regression Trees) for regression adjustment in estimating causal effects, followed by the extension of Hahn et al. (2020); Caron et al. (2022b), and the seminal work of Alaa & van der Schaar (2017; 2018) who first proposed multitask Gaussian Processes for causal effects estimation tasks, but limited to contexts with binary actions and a single (continuous) outcome. All these works highlight the advantages of Bayesian nonparametric models in terms of flexible non-linear function approximation and uncertainty quantification.

More specifically, the advantage of multitask GPs and DKL for causal learning lies in their sample efficiency gains. As observational data often feature imbalanced action arms with relatively few instances, sample splitting can result in under or over-fitting of the surfaces of interest. Multitask GPs and DKL both allow for information sharing when learning actions and outcomes correlated tasks, while only DKL guarantees better scalability and also makes the choice of a GP kernel less challenging, as the deep learning structure can itself learn arbitrarily complex non-linear functions (Wilson et al., 2016).

## 2 Problem Framework

In order to introduce the problems of causal effects identification and estimation, we borrow the notation from *do*-calculus and Structural Causal Models (SCM) (Pearl, 2009), and start by defining the latter:

**Definition 2.1** (SCM)**.** A Structural Causal Model is a 4-tuple $\langle \mathcal{E}, \mathcal{V}, F, p(\varepsilon) \rangle$ consisting of (subscript $j$ indicates a random element in the set):

---

[1] Full code at: `https://github.com/albicaron/CounterDKL`

1. $\mathcal{E}$: denoting a set of exogenous variables, defined as variables determined outside of the model.

2. $\mathcal{V}$: denoting a set of endogenous variables, defined as variables determined inside the model.

3. $F$: a set of functions $f_j \in F$ mapping each element $\varepsilon_j \in \mathcal{E}$ and every parent variables of $V_j \in \mathcal{V}$ — which we denote by $pa(V_j)$ — to the endogenous variables $V_j \in \mathcal{V}$, $f_j : \varepsilon_j \cup pa(V_j) \mapsto V_j$.

4. $p(\varepsilon_j)$: a probability distribution over $\varepsilon_j \in \mathcal{U}$.

A SCM implies a causal Directed Acyclic Graph (DAG), where nodes in the graph depict variables, i.e. $\varepsilon_j$ and $V_j$, while edges denote the functional causal relationships $f_j \in F$. In the following paragraphs, we examine the problem of identifying and estimating these causal relationships $f_j \in F$, under observational data, relying on SCMs and causal DAGs.

Suppose we have access to an observational dataset $\mathbb{D}_i = \{\boldsymbol{X}_i, A_i, \boldsymbol{Y}_i\} \sim p(\cdot)$, with $i \in \{1, ..., N\}$, where $\boldsymbol{X}_i \in \mathcal{X}$ is a set of covariates, $A_i \in \mathcal{A}$ a set of discrete actions[2] (or manipulative variables), and $\boldsymbol{Y}_i \in \mathbb{R}^M$ a set of $M$ different outcomes. Here we consider continuous type of outcomes for simplicity, but we extend the methods also to discrete outcomes (Milios et al., 2018), as in the experimental Section 5.2. The overarching goal is to identify and estimate the (average) effects of intervening on the manipulative variable $A_i$, by setting it equal to some value $a$, on the outcomes $\boldsymbol{Y}_i$. We denote the main quantity of interest, the joint interventional probability distribution of all the outcomes conditional on $A_i = a$ by using the *do*-operator as $p(\boldsymbol{Y}|do(A = a))$. We assume that the SCM is fully described by the following set of equations:

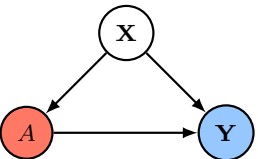

Figure 1: Causal DAG depicting the setting of interest, where arrows represent causal relationships. Set of observable covariates $\boldsymbol{X} \in \mathcal{X}$ satisfies the *backdoor criterion*, in that conditioning on them allows for identification of the causal effects of action $A \in \mathcal{A}$ on the outcomes $\boldsymbol{Y} \in \mathcal{Y}$.

$$
\begin{aligned}
\boldsymbol{X}_i &= f_X(\varepsilon_{i,X}), \\
A_i &= f_A\big(pa(A_i), \varepsilon_{i,A}\big) = f_A(\boldsymbol{X}_i, \varepsilon_{i,A}) \\
\boldsymbol{Y}_i &= \boldsymbol{f}_Y\big(pa(\boldsymbol{Y}_i), \varepsilon_{i,Y}\big) = \boldsymbol{f}_Y(\boldsymbol{X}_i, A_i, \varepsilon_{i,Y}) \,,
\end{aligned}
\tag{1}
$$

where: $\boldsymbol{f}_Y(\cdot)$ is a vector-valued function if multiple outcomes are considered; $pa(A_i)$ denotes parent variables (or causes) of $A_i$; $\varepsilon_{i,j}$ are error terms with a distribution $p(\varepsilon_{i,j})$. The equivalent causal DAG is represented in Figure 1, although we note that the methods presented in this work extend to other types of causal DAGs. We make two standard assumption for identification of the causal effect $A \rightarrow \boldsymbol{Y}$ in this scenario. The first is *unconfoundedness*, stating that there are no unobserved confounders, or equivalently that the set of observed covariates $\boldsymbol{X}_i \in \mathcal{X}$ is causally sufficient, in the sense that conditioning on $\boldsymbol{X}_i$ allows to identify the causal association between $A_i$ and $\boldsymbol{Y}_i$. Using Pearl's terminology, we equivalently say that $\boldsymbol{X}$ satisfies the *backdoor criterion* (see A.1 in the appendix for a formal definition), in that it "blocks all backdoor paths" from $A$ to all the outcomes $\boldsymbol{Y}$ (Pearl, 2009). The second assumption is *overlap* (also known as *positivity* or *common support* assumption). Overlap requires that $0 < p(A_i = a|\boldsymbol{X}_i = x) < 1$, $\forall a \in \mathcal{A}$ - i.e. that the observed actions allocation given $\boldsymbol{X}_i = \boldsymbol{x}$ is never deterministic, so we could theoretically observe data points for which $\boldsymbol{X}_i = \boldsymbol{x}$ in each of the discrete arms of $\mathcal{A}$ (A.2 in the appendix). This ensures that we have comparable units in terms of $\boldsymbol{X}_i$ in each action arm, so we can approximate $\boldsymbol{f}_Y(\boldsymbol{X}_i, A_i = a, \varepsilon_{i,Y})$ well enough. Violation of overlap for portions of $\mathcal{X}$ undermines generalization and extrapolation of model's prediction in those regions; thus, one must be careful as to which subpopulation, defined by a common support $\mathcal{X}_{\text{over}} \subseteq \mathcal{X}$, to target to estimate causal effects. Under these two assumptions, the multivariate interventional distribution $p(\boldsymbol{Y} \mid do(A = a), \boldsymbol{X} = \boldsymbol{x})$ can be recovered via *backdoor adjustment* as $p(\boldsymbol{Y} \mid do(A = a), \boldsymbol{X} = \boldsymbol{x}) = p(\boldsymbol{Y} \mid A = a, \boldsymbol{X} = \boldsymbol{x})$ (Pearl, 2009) (see Appendix A). Hence, provided that the covariates $\boldsymbol{X}_i$, direct common causes of $A_i$ and $\boldsymbol{Y}_i$, satisfy the backdoor criterion, we can estimate unbiased causal effects with the observed quantities in $\mathbb{D}_i = \{\boldsymbol{X}_i, A_i, \boldsymbol{Y}_i\}$.

---

[2]Extensions to settings with continuous action space $\mathcal{A} \subset \mathbb{R}$ (entailing a dose-response curve estimation) are non-trivial (Imai & van Dyk, 2004), and the multitask learning paradigm is not specifically suitable for them.

## 3 Counterfactual Learning with Multitask GPs

For ease of exposition, let us consider the simple case depicted by the causal DAG in Figure 1a, with a single continuous outcome $Y_i \in \mathbb{R}$, with $i \in \{1, ..., N\}$. We tackle the problem of estimating $p(Y|do(A = a))$ via non-linear regression-adjustment (Johansson et al., 2016; Shalit et al., 2017; Künzel et al., 2017; Nie & Wager, 2020; Caron et al., 2022a). In particular, we assume, in line with most of the previous works, additive noise structure[3], such that the outcome functional can be written as:

$$Y_i = f_Y(\boldsymbol{X}_i, A_i) + \varepsilon_{i,Y} \ , \quad \mathbb{E}(\varepsilon_{i,Y}) = 0 \,. \tag{2}$$

There are different ways in which one can derive an estimator for $p(Y|do(A = a))$ and its moments, e.g. $\mathbb{E}(Y|do(A = a))$, from (2) (Künzel et al., 2017; Caron et al., 2022a). Alaa & van der Schaar (2017; 2018) first proposed the use of multitask learning via Gaussian Process regression, in the specific context of conditional average treatment effects estimation, which is defined, assuming binary $A_i \in \{0, 1\}$, as the quantity $\tau(\boldsymbol{x}_i) = \mathbb{E}[Y_i|do(A_i = 1), \boldsymbol{x}_i] - \mathbb{E}[Y_i|do(A_i = 0), \boldsymbol{x}_i]$. The idea behind causal multitask GPs is to view the $D = |\mathcal{A}|$ interventional quantities $Y_a$, where $D$ is the number of discrete action arms, as the output from a vector-valued function $\boldsymbol{f}_Y(\cdot) : \mathcal{X} \mapsto \mathbb{R}^D$ (plus noise), modelled with a GP prior:

$$\boldsymbol{f}_Y(\cdot) \sim \mathcal{GP}\Big(\boldsymbol{m}(\cdot), K(\cdot, \cdot)\Big) \,, \tag{3}$$

with mean $\boldsymbol{m}(\boldsymbol{x}_i) \in \mathbb{R}^D$ and covariance/kernel function $K(\boldsymbol{x}_i, \boldsymbol{x}_j) \in \mathbb{R}^D \times \mathbb{R}^D$, given two $P$-dimensional input points $\boldsymbol{x}_i, \boldsymbol{x}_j \in \mathcal{X}$ for units $i$ and $j$. Given the likelihood function as a multivariate Gaussian $p(\boldsymbol{y}_i|\boldsymbol{f}_Y, \boldsymbol{x}_i, \Sigma) \triangleq \mathcal{N}(\boldsymbol{f}_Y(\boldsymbol{x}_i), \Sigma)$, where $\Sigma \in \mathbb{R}^D \times \mathbb{R}^D$ is the error covariance diagonal matrix with $\{\sigma_a^2\}_{a=1}^D$ on the diagonal and $\boldsymbol{y}_i \in \mathbb{R}^D$ an output point, the posterior predictive distribution for a train set covariate realization $\boldsymbol{x}_i \in \mathcal{X}$, train set outcome realization $\boldsymbol{y}_i \in \mathbb{R}$ and a test set covariate realization $\boldsymbol{x}_j^* \in \mathcal{X}$ is obtained as, assuming zero prior mean $\boldsymbol{m}(\cdot) = \boldsymbol{0}$ for simplicity:

$$p\big(\boldsymbol{f}_Y(\boldsymbol{x}_j^*) \mid (\boldsymbol{x}_i, \boldsymbol{y}_i), \boldsymbol{f}_Y, \phi\big) \triangleq \mathcal{N}\big(\boldsymbol{f}_Y^*(\boldsymbol{x}_j^*), K^*(\boldsymbol{x}_j^*, \boldsymbol{x}_j^*)\big) \,,$$
$$\boldsymbol{f}_Y^*(\boldsymbol{x}_j^*) = K(\boldsymbol{x}_j^*, \boldsymbol{x}_i)H\boldsymbol{y} \,, \quad K^*(\boldsymbol{x}_j^*, \boldsymbol{x}_j^*) = K(\boldsymbol{x}_j^*, \boldsymbol{x}_j^*) - K(\boldsymbol{x}_j^*, \boldsymbol{x}_i)HK^\top(\boldsymbol{x}_j^*, \boldsymbol{x}_i) \,, \tag{4}$$
$$\text{where} \quad H = \Big[K(\boldsymbol{x}_i, \boldsymbol{x}_i) + \Sigma\Big]^{-1} \,,$$

and where $\phi$ denotes the model parameters and $\boldsymbol{f}_Y^*(\boldsymbol{x}_j^*)$ the function evaluated at a test point $\boldsymbol{x}_j^*$. Under zero prior mean $\boldsymbol{m}(\cdot) = \boldsymbol{0}$, the multitask GP in (3) is fully parametrized by its kernel function $K(\cdot, \cdot)$. The structure of the kernel function in a multitask GP is what induces task-relatedness when fitting the multi-valued surface $\boldsymbol{f}_Y(\cdot)$.

### 3.1 The multitask kernel

The simplest specification for the multitask kernel matrix is given by the *separable kernels* structure, which assumes single entries in $K(\cdot, \cdot)$ to be of the form $k_{a,a'}(\boldsymbol{x}_i, \boldsymbol{x}_j) = k(\boldsymbol{x}_i, \boldsymbol{x}_j)\,k_A(a, a') = k(\boldsymbol{x}_i, \boldsymbol{x}_j)\,b_{a,a'}$, with action $a \in \mathcal{A} = \{1, ..., D\}$. Here, $k(\boldsymbol{x}_i, \boldsymbol{x}_j)$ represents a base kernel (e.g. linear, squared exponential, Matérn, etc.) while $b_{a,a'} = k_A(a, a')$ is the generic entry of the $D \times D$ **coregionalization matrix** $B$, which contains the parameters governing task-relatedness over the actions $A$. In the trivial case where $b_{a,a'} = 0$ we have that tasks $a$ and $a'$ are uncorrelated, i.e. actions $a$ and $a'$ are unrelated in the way they affect the outcome $Y$.

A slightly more general framework, which we adopt in this work, is given by the *sum of separable kernels* structure (Álvarez et al., 2012). This assumes that the single entry of $K(\cdot, \cdot)$ reads $k_{a,a'}(\boldsymbol{x}_i, \boldsymbol{x}_j) = \sum_{q=1}^Q B_q\,k_q(\boldsymbol{x}_i, \boldsymbol{x}_j)$, i.e. the sum of $Q$ different coregionalization matrices $B_q$ with associated base kernel $k_q(\cdot, \cdot)$. In compact matrix notation this translates into $K(X, X') = \sum_{q=1}^Q B_q \otimes K_q(X, X')$ for two different input matrices $X, X' \in \mathcal{X}$, where $\otimes$ is the Kronecker product. Imposing a *sum of separable kernels* structure is equivalent to assuming that the collection of action-specific functions $\{f_{Y_a}(\cdot)\}_{a=1}^D$ generates from $Q \leq D$

---

[3]Technically this makes the setup semi-parametric, rather than fully nonparametric.

common underlying independent latent GP functions $\{u_q(\cdot)\}_{q=1}^Q$, parametrized by their base kernel $k_q(\cdot, \cdot)$, that is $\text{cov}(u_q(\boldsymbol{x}_i), u_{q'}(\boldsymbol{x}_j)) = k_q(\boldsymbol{x}_i, \boldsymbol{x}_j)$ (Álvarez et al., 2012).

In terms of the form of the coregionalization matrices $B_q$, with $q \in \{1, ..., Q\}$, we will follow the **linear model of coregionalization** (LMC), which assumes that each $B_q$ is equal to $B_q = L_q L_q'$, with single entries $b_{q;(a,a')} = \sum_{r=1}^{R_q} \alpha_{a,q}^r \alpha_{a',q}^r$. $R_q$ represents the number of GP samples obtained from the same latent GP function $q$, $u_q(\cdot)$. Thus, adopting the LMC for causal learning is equivalent to assuming that correlation in the $\{f_{Y_a}(\cdot)\}_{a=1}^D$ action-specific functions, modelled through the multitask kernel $K(\cdot, \cdot)$, arises from $Q$ different samples of $R_q$ GP functions with the same kernel $k_q(\cdot, \cdot)$, drawn from $Q \leq D$ different independent latent GP processes $\{u_q(\cdot)\}_{q=1}^Q$. To express this more compactly, a causal multitask GP model under the LMC reads $\boldsymbol{f}_Y(\cdot) \sim \mathcal{GP}(\boldsymbol{m}(\cdot), K(\cdot, \cdot))$, with single entries of $K(\cdot, \cdot)$ being

$$
k_{a,a'}(\boldsymbol{x}_i, \boldsymbol{x}_j) = \sum_{q=1}^Q B_q \, k_q(\boldsymbol{x}_i, \boldsymbol{x}_j) = \sum_{q=1}^Q A_q A_q' \, k_q(\boldsymbol{x}_i, \boldsymbol{x}_j), \quad \text{implying}
$$

$$
\text{cov}(f_a(\boldsymbol{x}_i), f_{a'}(\boldsymbol{x}_j)) = \sum_{q,q'}^Q \sum_{r,r'}^{R_q} \alpha_{a,q}^r \alpha_{a',q'}^{r'} \, \text{cov}(u_q^r(\boldsymbol{x}_i), u_{q'}^{r'}(\boldsymbol{x}_j)). \tag{5}
$$

In our specific case, as in Alaa & van der Schaar (2018), we employ a special case of LMC, named **intrinsic coregionalization model** (ICM) (Bonilla et al., 2008), where the underlying latent GP function is unique ($Q = 1$), so that $k_{a,a'}(\boldsymbol{x}_i, \boldsymbol{x}_j) = B \, \tilde{K}(\boldsymbol{x}_i, \boldsymbol{x}_i')$, with unique base kernel $\tilde{K}(\cdot, \cdot)$. The ICM specification attempts to avoid severe parameter proliferation in high-dimensional settings with multiple correlated actions $D = |\mathcal{A}|$, while still being capable of capturing task-relatedness through the relatively simple structure of $B$. However, beside the issue of parameter proliferation when $\mathcal{A}$ features multiple discrete actions, exact GP regression is also known to scale poorly with sample size and cardinality of input space $|\mathcal{X}|$, and direct likelihood maximization methods face issues in over-parametrized models, although some solutions, such as variational methods (Titsias, 2009; Hensman et al., 2013), might be adopted for better scalability.

## 3.2 Why multitask counterfactual learning?

We know that asymptotically, under no sample selection bias, the best approach to estimate the causal quantities $\boldsymbol{f}_Y(\boldsymbol{x}_i)$ would be a "T-Learner" (Künzel et al., 2017; Caron et al., 2022a), which implies splitting the sample and fitting separate models for each arm $A = a$, $\hat{f}(\mathbf{x})_a$. However, in finite samples flawed by selection bias, which is the typical case of observational studies, this is often not the best strategy (Hahn et al., 2020; Caron et al., 2022a). By simply extending the result in Alaa & van der Schaar (2018), we hereby show how increasing action and covariates spaces make the problem of estimating causal effects harder in terms of minimax risk, i.e. in terms of optimal convergence rates for any nonparametric estimator for a given space of functions. We particularly consider the problem of estimating Conditional Average Treatment Effects (CATE) between action $a$ and $b$, defined under the SCM specified by (2) as $\tau_{a,b}(\boldsymbol{x}_i) = \mathbb{E}[Y_i | do(A_i = a), \boldsymbol{X}_i = \boldsymbol{x}] - \mathbb{E}[Y_i | do(A_i = b), \boldsymbol{X}_i = \boldsymbol{x}] = f_a(\boldsymbol{x}_i) - f_b(\boldsymbol{x}_i)$. We assume that all the $f_a$, $\forall a \in \mathcal{A}$ belong to the Hölder ball class of $\alpha_a$-smooth functions $\mathcal{H}(\alpha_a)$, with $\alpha_a - 1$ bounded derivatives and bounded in sup-norm by a constant $C > 0$. Considering an $L^2$-norm loss function on $\tau_{a,b}(\boldsymbol{x}_i)$, namely $\mathbb{E}[\|\hat{\tau}_{a,b}(\boldsymbol{x}_i) - \tau_{a,b}(\boldsymbol{x}_i)\|_{L^2}^2]$, the difficulty of CATE estimation with a nonparametric model $\psi \in \Psi$ can be specified by the optimal minimax rate of convergence, that we define as follows (proof in the Appendix A section).

**Corollary 3.1** (Minimax rate). *Assume covariate space is $\mathcal{X} = [0, 1]^P$, and $f_a$ depends on $P_a$ covariates s.t. $P_a \leq P$, $\forall a \in \mathcal{A}$. Define $n_{a,b} < N$ as the subsample identified by action $a$ and $b$. If both $f_a \in \mathcal{H}(\alpha_a)$ and $f_b \in \mathcal{H}(\alpha_b)$, then CATE optimal minimax rate for a $L^2$-norm loss function is:*

$$
\inf_{\hat{\tau}_\psi} \sup_{f_a, f_b} \mathbb{E}[\|\hat{\tau}_{a,b}(\boldsymbol{x}_i) - \tau_{a,b}(\boldsymbol{x}_i)\|_{L^2}^2] \asymp n_{a,b}^{-\left(1 + \frac{1}{2}\left(\frac{P_a}{\alpha_a} \vee \frac{P_b}{\alpha_b}\right)\right)^{-1}} \vee \log\left(\frac{P^{P_a + P_b}}{P_a^{P_a} P_b^{P_b}}\right)^{\frac{1}{n_{a,b}}}, \tag{6}
$$

*where $x \vee y = \max\{x, y\}$ and $\asymp$ is asymptotical equivalence.* The first terms on the RHS of 6 relates to the problem of CATE function approximation, while the second term to the degree of sparsity of the CATE. Thus,

the optimal minimax rate, which minimizes the loss in the worst case scenario permitted, is asymptotically as complex as the hardest of these two tasks. The "tightness" of this rate depends on the cardinality of the predictor space $P$ and of the subsample defined by action $a$ and $b$. This implies that, in the presence of multiple discrete actions (where some arms are likely to be very imbalanced), the rate will inevitably grows larger. For these reasons, a multitask GP prior over the actions is well suited to tackle selection bias and particularly estimation in regions with poor overlap, i.e. regions in $\mathcal{X}$ where we mainly observe data points with specific action $A_i = a$ and very few others. In addition to this, as shown by Alaa & van der Schaar (2018), a multitask GP prior can achieve the optimal minimax rate of Corollary 3.1 in its posterior consistency.

Hence, splitting the sample into $n_a$ subgroups and fitting independent models can be very sample inefficient in these settings (6). Multitask GPs can aid extrapolation in such cases of strong sample selection bias, by learning the correlated functions $\{f_{Y_a}(\cdot)\}_{a=1}^D$ jointly as $\boldsymbol{f}_Y(\cdot)$. The first row plots in Figure 2 provide a very simple one-covariate example of how multitask learning addresses the issue of extrapolation and prediction in poor overlap regions. Fitting the two surfaces $f_{Y1}(x_i)$ and $f_{Y0}(x_i)$ (dashed lines) through separate GP regressions results in a bad fit out of overlap regions (top-left plot) in this specific case. Multitask coregionalized GP attempts to fix this problem by embedding the assumption that the two surfaces share similar patterns via joint estimation of $\boldsymbol{f}_{Y_a}(x_i)$ and their task-relatedness parameters, increasing sample efficiency (right panel). When the two surfaces share minor patterns instead, such as in the second row plots of Figure 2, the sample efficiency gains are less significant; and in some more extreme cases where the surfaces do not share any similar pattern at all, assuming a multitask GP prior might also introduce bias. The issue of partial overlap might be less severe in scenarios with larger sample size, but not always; however, in settings with strong sample selection bias, or settings with multiple discrete actions or action spaces that grow with the sample size, the issue remains relevant. This is because the sampling mechanism is inherently flawed, so that even with infinite samples, poor overlap regions do not fade away, independently of the modelling assumptions one makes.

Modelling assumptions on how to estimate $\{f_{Y_a}(\cdot)\}_{a=1}^D$ hence ultimately depends on domain expert knowledge about the setting Hahn et al. (2020); Caron et al. (2022c). If one possesses prior knowledge that the causal effects might be fairly homogeneous across units with different covariate realizations $\boldsymbol{X}_i = \boldsymbol{x}_i$, so that the CATE function is likely to display simple patterns (first row, Figure 2), then using a multitask approach would actually help incorporate this assumption in the model. Conversely, if one believes instead that CATE is likely to be a rather complex function itself (second row, Figure 2), estimating $\{f_{Y_a}(\cdot)\}_{a=1}^D$ independently would be a better choice.

### 3.3 Multiple Output Designs

Reverting back to setups with multiple correlated outcomes, we introduce a simple extension to the class of counterfactual GPs presented above that involves an extra multitask learning layer over the $M$ outcomes $\boldsymbol{Y}_i$, in addition to the one over $A_i$. The way we formulate this is through an additional coregionalization matrix, in the same fashion as we did for the actions case. This extended version featuring *stacked coregionalization*, has a GP prior of the following form:

$$\boldsymbol{Y}_i = \boldsymbol{f}_{Y_a}(\boldsymbol{x}_i) + \boldsymbol{\varepsilon}_i \ , \quad \mathbb{E}(\boldsymbol{\varepsilon}_i) = \boldsymbol{0}$$

$$\boldsymbol{f}_{Y_a}(\cdot) \sim \mathcal{GP}\big(\boldsymbol{0}, K_Y(\cdot, \cdot)\big) \ , \quad K_Y(\cdot, \cdot) = B_Y \otimes B_A \otimes \tilde{K}(\cdot, \cdot) \ , \tag{7}$$

where $B_Y$ is the $M \times M$ coregionalization matrix over the outcomes, $B_A$ the $D \times D$ coregionalization matrix over the actions and $\tilde{K}(\cdot, \cdot)$ is the base kernel. The vector-valued function $\boldsymbol{f}_{Y_a}(\cdot)$ in this case includes all the single-valued functionals $\{f_{a,m}(\cdot)\}_{a,m}^{D,M}$. The extra multitask learning layer over the outcomes $\boldsymbol{Y}_i$ is aimed at increasing sample efficiency by borrowing information among correlated outcomes, as opposed to fitting $M$ separate counterfactual GPs with a single coregionalization layer over $A$, but it is also conceptually sound, as the quantity of interest is indeed the joint interventional distribution $p(\boldsymbol{Y}|do(A = a), \boldsymbol{X} = \boldsymbol{x})$, which accounts for and explicitly models correlation between the outcomes, rather than the collection of marginal distributions $\{p(Y_m|do(A = a))\}_{m=1}^M$, which leaves correlation unspecified. As we will address in the later

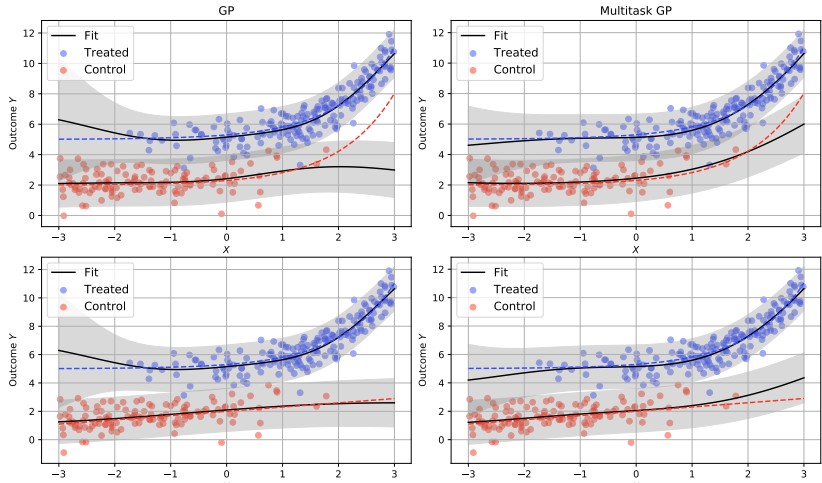

Figure 2: Simple one covariate example, with $\mathcal{A} = \{0, 1\}$. Overlap is guaranteed to hold over the whole support $\mathcal{X}$ in the data generating process, i.e. every unit has non-zero probability of being assigned to either $A_i = 1$ or $A_i = 0$, but $p(A_i = 1|X_i)$ is generated as an increasing function of $X_i$ (selection bias). In the top row simulation, the two underlying counterfactual surfaces $f_{Y_d}(x_i)$ (dashed lines) are generated with very similar patterns, thus GP (left panel) is unable to borrow information from the other arm in poor overlap regions contrary to multitask GP (right panel). In the bottom row simulation instead we generate less similar surfaces, so borrowing of information through multitask GP does not lead to any improvement.

section, although the extra layer defined by $B_Y$ allows for higher sample efficiency, it also poses some issues due to parameter proliferation and stability of the optimization problem in high dimensions.

## 4 Counterfactual Multitask Deep Kernel Learning

Gaussian Processes regressions are known to scale poorly with high dimensions. Their typical computational cost amounts to $\mathcal{O}(n^3)$ for training points and $\mathcal{O}(n^2)$ for test points. Similarly, coregionalized GPs suffer from over-parametrization and instability in the optimization procedure as the number of inputs $P$ and the number of discrete actions $D$ increase. Deep Kernel Learning (DKL) was firstly introduced by Wilson et al. (2016) with the aim of combining the scalability of Deep Learning methods and the nonparametric Bayesian uncertainty quantification of GPs in tackling prediction tasks in high-dimensional settings. Given a base kernel $k(\cdot, \cdot)$ (e.g. linear, squared exponential, etc.), a DKL structure learns a functional $f_{Y_a}(\cdot)$ by passing the $P$ inputs $\boldsymbol{X}_i \in \mathcal{X}$ through a deep architecture (a fully-connected feedforward neural network in our case), which maps them to a lower dimensional representation space via non-linear activation functions. The base kernel $k(\cdot, \cdot)$ is then applied in this lower dimensional representation space, $k(h^{(l)}, h^{(l)'})$, where $h^{(l)}$ is a neural network's hidden layer, constituting a final Gaussian Process layer (or an infinite basis functions representation layer). The resulting mathematical object can be described as a kernel being applied to a concatenation of linear and non-linear functions of the inputs, namely $\tilde{K}(\boldsymbol{x}, \boldsymbol{x}') = K(g_1 \circ \dots \circ \dots \circ g_l(\boldsymbol{x}), g_1 \circ \dots \circ g_l(\boldsymbol{x}'))$ (Bohn et al., 2019). Thus, the DKL architecture is end-to-end, fully-connected and learnt jointly: the $P$ inputs are passed on to $\ell$ hidden neural nets layers where the last hidden layer before the GP layer typically maps them to a lower dimensional representation space (with e.g. two hidden units). This is what generates benefits in terms of scalability compared to a classic GP, as the base kernel $k(\cdot, \cdot)$ is applied to a lower dimensional representation space, rather than the higher dimensional inputs space directly. Another intrinsic advantage of DKL is that the deep architecture preceeding the GP layer can itself learn arbitrarily complex function, so the choice of a specific GP kernel becomes less cumbersome. For example, Wilson et al. (2016) show that DKL is more robust in recovering step functions, due to weaker smoothness assumptions compared to standard GP kernels.

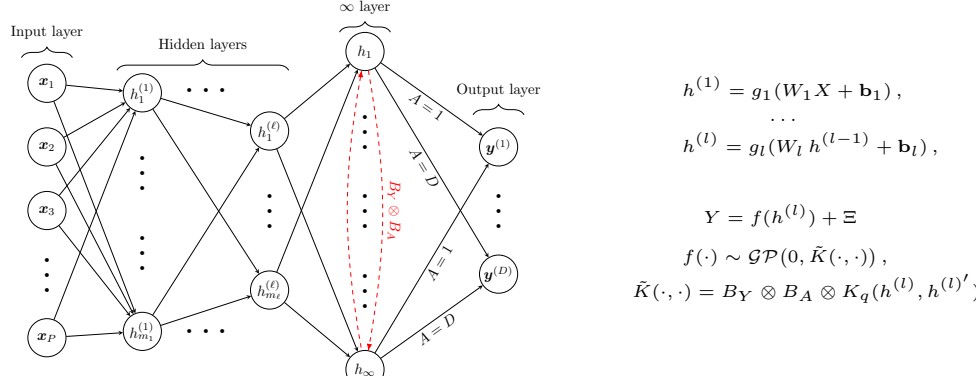

Figure 3: Counterfactual multitask DKL architecture. The $P$ raw inputs are passed through a deep learning structure with $\ell$ hidden layers. Multioutput separable kernels (inducing coregionalization over actions $A$ and outcomes $\boldsymbol{Y}$) are then applied to the last Gaussian Process hidden layer, before the $M$ action-specific output layer. Parameters are estimated jointly by minimizing the negative log likelihood.

DKL naturally presents some limitations concerning the more burdensome parameter tuning (e.g., hidden layers and units selection) and the fact that they more easily tend to overfit when overly-parametrized (we refer to Ober et al. (2021) for a more detailed discussion of the issue). The kernel $k(\cdot, \cdot)$ in the last GP layer of a DKL architecture can easily incorporate the separable kernel structure for multitask learning, in the same fashion as the class of causal GPs presented earlier. Thus, we propose a multitask modelling framework to induce correlation across the action-specific functions $\{f_{Y_a}(\cdot)\}_{a=1}^{D}$, under the name of Counterfactual DKL (CounterDKL), where a similar Intrinsic Coregionalization Model (ICM) in the same spirit of (5) is placed on the last hidden layer of the neural network $h^{(l)} = g_l(W_l\, h^{(l-1)} + \boldsymbol{b}_l)$, where $(W_l, \boldsymbol{b}_l)$ are the last layer's weights and bias, such that

$$\boldsymbol{f}_{Y_a}(h^{(l)}) \sim \mathcal{GP}\Big(0, K(\cdot, \cdot)\Big), \quad \text{where}$$
$$K(h^{(l)}, h^{(l)'}) = K_A(a, a') \otimes \tilde{K}(h^{(l)}, h^{(l)'}) = B \otimes \tilde{K}(h^{(l)}, h^{(l)'})$$

(8)

In this case, the Kronecker product of the coregionalization matrix occurs in the last hidden layer, and features lower dimensional representations instead of the potentially large number of raw inputs. Similarly, we can induce coregionalization over the $M$ outcomes by adding another level of coregionalization, with the kernel reading $K(h^{(l)}, h^{(l)'}) = K_Y(y, y') \otimes K_A(a, a') \otimes K_q(h^{(l)}, h^{(l)'}) = B_Y \otimes B_A \otimes K_q(h^{(l)}, h^{(l)'})$. Figure 3 graphically depicts a counterfactual multitask Deep Learning architecture, with fully-connected hidden layers, a final (infinite) GP layer and the $M$ action-specific outcomes. The parameter space in CounterDKL comprises: i) the set of deep neural network's weight matrices $\{W_i\}_{i=1}^{l}$ and biases $\{\boldsymbol{b}_i\}_{i=1}^{l}$; ii) the base kernel $\tilde{K}$ hyperparameters $\Phi$, e.g. in the case of squared exponential kernel $\Phi$ includes lengthscales and variances parameters $\Phi = \{\boldsymbol{\ell}, \sigma^2\}$; iii) the coregionalization matrix $B$ entries. Hence, the parameter space is the collection $\Theta = (\{W_i\}_{i=1}^{l}, \{\boldsymbol{b}_i\}_{i=1}^{l}, \Phi, B)$. These parameters are estimated jointly via maximization of the log-marginal likelihood. More details are provided in the Appendix Section D and in Wilson et al. (2016); Gardner et al. (2018). In the next section, we will investigate properties of counterfactual multitask GPs and DKL on a variety of experiments.

## 5 Experiments

The fundamental problem of causal inference is that the interventional quantity $p(\boldsymbol{Y}|do(A = a), \boldsymbol{X}_i = \boldsymbol{x}_i)$ is never observable, so we have to resort to simulation to fully evaluate the methods on individual causal effects (ICE) estimation. We evaluate the performance of counterfactual GPs and counterfactual DKL on a data generating process with three different tasks, and on a real-world example combining experimental and observational data. For the first simulated experiment, we construct the DGP such that the *backdoor*

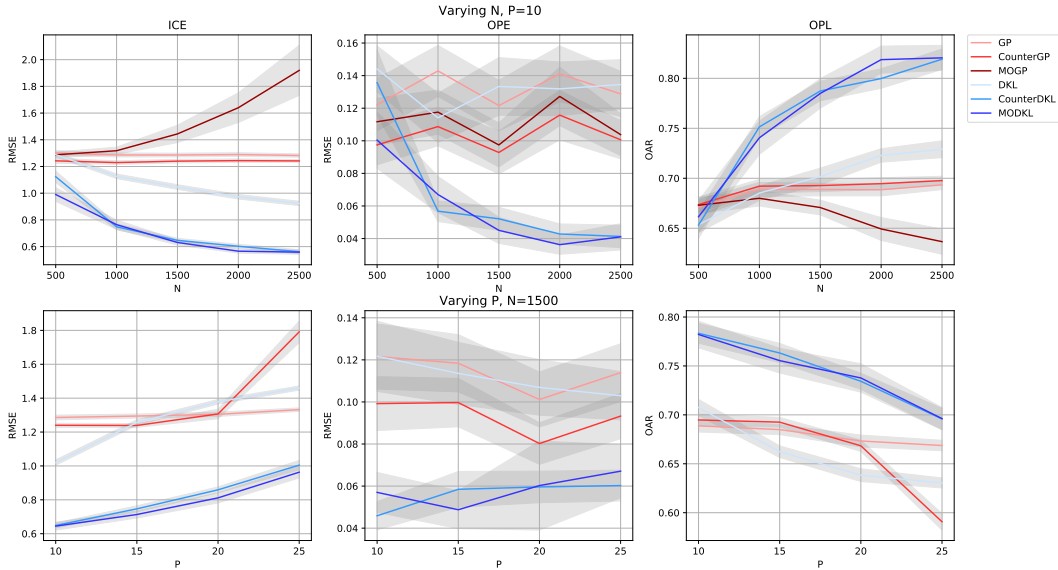

Figure 4: Results on performance of the methods compared, in terms of RMSE or Optimal Allocation Rate (OAR), averaged across $B = 100$ replications for each $N \in \{500, 1000, 1500, 2000, 2500\}$ (first row) and each $P \in \{10, 15, 20, 25\}$ (second row). First column: RMSE evaluated on the individual causal effect (ICE) estimation task (on the test set). Second column: RMSE evaluated on the OPE task. Third column: OAR on the OPL task, defined as percentage of units correctly allocated to the best action among the $D$ ones.

*criterion* holds for $\boldsymbol{X}_i \in \mathcal{X}$. The GPs and DKL implementations in the simulated examples all make use of the KISS-GP approximation to compute the base kernel covariance matrix as $K_q = M K_{U,U}^{\text{deep}} M^\top$ in the GP layer for better scalability (Wilson & Nickisch, 2015; Wilson et al., 2016)[4].

## 5.1  Simulated Example

We consider a simulated setting with $D = 4$ possible actions $\mathcal{A} = \{0, 1, 2, 3\}$ and $M = 2$ correlated outcomes $\boldsymbol{Y} = (Y_1, Y_2) \in \mathbb{R}^2$. Actions and outcomes are generated according to a policy $\pi_b(\boldsymbol{x}_i) = p(A_i = a | \boldsymbol{X}_i = \boldsymbol{x}_i)$ and an outcome function $\boldsymbol{f}_Y(\boldsymbol{x}_i)$, both dependent on the covariates $X_i \in \mathcal{X}$. The probabilistic DGP is fully described in the Appendix B of supplementary materials. The models we compare are the following: i) separate standard GP regressions, employed to fit $f_{Y_d}(\cdot)$ distinctly for each outcome and for each action (**GP**); ii) counterfactual multitask GP regression (Alaa & van der Schaar, 2017; 2018), with coregionalization over $A_i$ only, meaning that we fit two separate models for each outcome, but a unique multi-valued function model for $A_i$ (**CounterGP**); iii) counterfactual multioutput GP regression, a unique model with coregionalization both over $A_i$ and $Y_i$ (**MOGP**); iv) separate DKL regressions with 3 hidden layers of $[50, 50, 2]$ units, the equivalent of i) but with deep kernel implementation (**DKL**); v) counterfactual multitask DKL regression with 3 hidden layers of $[50, 50, 2]$ units, the DKL equivalent of ii) (**CounterDKL**); vi) counterfactual multioutput DKL, the DKL equivalent of iii) (**MODKL**). In particular, we consider two slightly different versions of this setup. In the first version we fix the number of covariates to $P = 10$ (only 7 of them being relevant for the estimation) and study the behaviour of the estimators with increasing sample size $N \in \{500, 1000, 1500, 2000, 2500\}$. In the second version we fix sample size to $N = 1500$ and study the behaviour of the estimators with increasing number of covariates $P \in \{10, 15, 20, 25\}$. Performance of the models is evaluated on the following three related tasks:

- **ICE**: The first is the prediction of Individual Causal Effects (ICE). This tackles the estimation of the average causal effect of playing action $A_i = a$ on outcome $Y_i$, given a certain realization of the covariates

---

[4] All experiments were run on a Intel(R) Core(TM) i7-7500U CPU @ 2.70GHz, 8Gb RAM CPU.

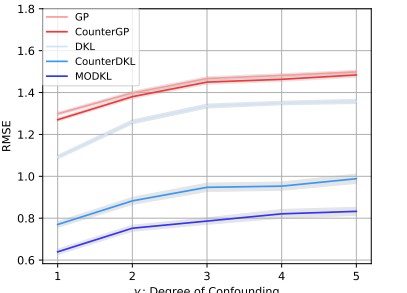 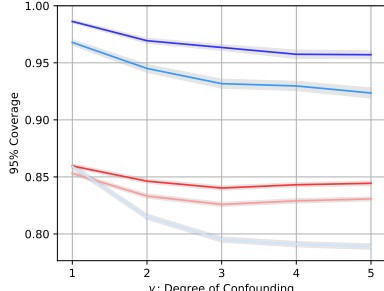

Figure 5: Models' performance in terms of RMSE (left plot) and 95% Coverage (right plot), in estimating Individual Causal Effects (ICE) on a 20% left-out test set, given an increasing level of confounding, represented by the $\gamma$ parameter: higher values of $\gamma$ corresponds to higher probability of being assigned to one of the two action $A_i = \{3, 4\}$, thus generating more arms imbalance.

space, $\boldsymbol{X}_i = \boldsymbol{x}$, i.e. the estimation of `ICE`: $\mathbb{E}(Y_i | do(A_i = a), \boldsymbol{X}_i = \boldsymbol{x}_i)$. This is carried out using a 80% training set, and evaluated via RMSE on a 20% left-out test set.

- **OPE**: The second is Off-Policy Evaluation, which is concerned with quantifying how good a given alternative policy $\pi_e$ is, compared to the action allocation policy that generated the data (behavior policy, $\pi_\beta$). This is done by estimating the *policy value*, defined as the cumulative reward $\mathcal{V}(\pi_e) = \mathbb{E}_{\mathcal{X}, \mathcal{A}, \mathcal{Y}}\left[\sum_i \pi_e(a_i | x_i)\left(\boldsymbol{Y}_i | do(A_i = a_i)\right)\right]$ originating from $\pi_e$. In our case we pick the alternative policy to be a uniformly-at-random action allocation, i.e. $\pi_e \sim \text{Multinom}(.25, .25, .25, .25)$. Performance is evaluated through RMSE on the entire sample.

- **OPL**: The last is Off-Policy Learning, which is concerned with finding the optimal policy $\pi^* : \mathcal{X} \to \mathcal{A}$, that is the policy that generates the highest *policy value*: $\pi_p^* \in \arg\max_{\pi_p \in \Pi} \mathcal{V}(\pi_p)$. This last task is evaluated through an Accuracy metric on the whole sample, which we label Optimal Allocation Rate (OAR), indicating the percentage of units correctly assigned to their specific optimal action $\pi^*(x_i) = a$, i.e. the action that generates the best outcome for them.

Since we are dealing with $M = 2$ outcomes, we produce performance measurements on RMSE and optimal allocation rate for both outcomes and then average them, assuming both outcomes are given equal policy importance and live on the same scale. For both versions of the setup, namely increasing $N$ and increasing $P$, we replicate the experiment $B = 100$ times to obtain Monte Carlo averages and 95% confidence intervals for the metrics. Results are depicted in Figure 4. RMSE performance in all models for increasing $N$ and $P$ behaves accordingly to Corollary 3.1. CounterDKL and MODKL perform consistently better than the GP models, as they scale better with an increasing sample size $N$ and increasing number of predictors $P$. Particularly MOGP's performance deteriorates for issues related to stability of the marginal likelihood maximization and over-parametrization, as we had to omit it from the study of increasing predictors due to failed convergence for $P > 10$. The advantages over standard DKL regression instead are entirely attributable to sample efficiency gains from multitask coregionalization in CounterDKL and MODKL, both in the increasing $N$ and increasing $P$ studies. In the case of increasing $P$, we emphasize that as predictor space grows larger the causal DGP becomes relatively sparser (only 7 predictors out of $P$ remain relevant for the estimation), especially in the case of $P = \{20, 25\}$. So in these two cases the batch of DKL models would perhaps achieve better performance from increasing the number of hidden units or hidden layers and adding regularization (dropout, $\ell 1$ or $\ell 2$ regularizers) in the deep architecture part.

In addition, we run a slightly different version of the ICE experiment above, to further investigate properties of the models in terms of uncertainty quantification, that we measure through the 95% coverage of each ICE

| Model | Train MAE | Test MAE | Train $\mathcal{R}_{\text{pol}}$ | Test $\mathcal{R}_{\text{pol}}$ | Runtime (s) |
|---|---|---|---|---|---|
| GP | $0.033 \pm 0.006$ | $0.036 \pm 0.008$ | $0.22 \pm 0.02$ | $0.27 \pm 0.02$ | $171.3 \pm 16.1$ |
| CounterGP | $0.033 \pm 0.006$ | $0.035 \pm 0.007$ | $0.24 \pm 0.01$ | $0.27 \pm 0.02$ | $248.6 \pm 6.4$ |
| PCA + GP | $0.073 \pm 0.002$ | $0.074 \pm 0.003$ | $0.22 \pm 0.01$ | $0.27 \pm 0.02$ | $66.3 \pm 2.4$ |
| PCA + CounterGP | $0.074 \pm 0.001$ | $0.074 \pm 0.001$ | $0.23 \pm 0.01$ | $0.26 \pm 0.02$ | $126.1 \pm 3.9$ |
| AutoEnc + GP | $0.075 \pm 0.004$ | $0.075 \pm 0.003$ | $0.21 \pm 0.03$ | $0.27 \pm 0.02$ | $76.0 \pm 3.0$ |
| AutoEnc + CounterGP | $0.076 \pm 0.003$ | $0.076 \pm 0.003$ | $0.24 \pm 0.02$ | $0.30 \pm 0.03$ | $138.7 \pm 9.2$ |
| DKL | $0.029 \pm 0.011$ | $0.042 \pm 0.015$ | $\mathbf{0.20 \pm 0.01}$ | $\mathbf{0.21 \pm 0.02}$ | $44.8 \pm 3.3$ |
| CounterDKL | $\mathbf{0.011 \pm 0.003}$ | $\mathbf{0.015 \pm 0.005}$ | $\mathbf{0.22 \pm 0.01}$ | $\mathbf{0.25 \pm 0.02}$ | $122.7 \pm 7.4$ |

Table 1: Train and test set performance on the Jobs data experiment in terms of Mean Absolute Error (MAE) in estimating ATT, Policy Risk ($\mathcal{R}_{\text{pol}}$) and overall runtime (s), with 10-fold cross-validated 95% intervals. Bold indicates best performance.

estimates (then averaged over actions and outcomes). This is defined as

$$\text{Coverage}_{95\%} = \frac{1}{N} \sum_{i=i}^{N} \mathbb{I}\Big(\hat{f}_a(\boldsymbol{x}_i)_{low} \leq f_a(\boldsymbol{x}_i) \leq \hat{f}_a(\boldsymbol{x}_i)_{upp}\Big),$$

where $\hat{f}_a(\boldsymbol{x}_i)_{low}$ and $\hat{f}_a(\boldsymbol{x}_i)_{upp}$ are the lower and upper bands of the 95% credible interval output by the model on $\hat{f}_a(\boldsymbol{x}_i)$, while $f_a(\boldsymbol{x}_i)$ is the true individual counterfactual outcome corresponding to action $A_i = a$. Given fixed $N = 2000$ and $P = 20$ and a similar data generating process as before, we introduce the parameter $\gamma$, which governs the level of confounding, or the degree of groups imbalance in terms of action allocation. Particularly, for increasing values of $\gamma$, we assign higher probability of choosing one of the two action $A_i = \{3, 4\}$. This generates action arms imbalance as it leaves gradually less units in arms $\{1, 2\}$. Results are gathered in Figure 5, where MODKL display higher performance both in terms of error and uncertainty quantification.

## 5.2 Real-World Example: Job Training Programs and Unemployment

We demonstrate the efficiency of CounterDKL also on a second experiment taken from Shalit et al. (2017), involving a popular real-world study on a job training program, dating back to LaLonde (1986). The distinctive feature of this dataset is that it combines a randomized and an observational subgroups, where the aim is to estimate the effects of participation on a job training program on earnings and employment. The randomized experiment features 297 treated and 425 control units; The observational subsample is instead made of 2490 control units only. The binary treatment $A_i \in \{0, 1\}$ denotes participation to the job training program. The original outcome $Y_i$ is earnings after the program, which is censored continuous ($Y_i = 0$ for unemployed units). However, following Shalit et al. (2017), we construct a binary indicator $Y_i \in \{0, 1\}$ denoting employment status at the end of the job training program as outcome. This gives us the opportunity to demonstrate the use of the methods presented in this paper also on binary/categorical type of outcomes. To this end we use the classification method for GPs proposed in Milios et al. (2018), where class labels are interpreted as coefficients of a degenerate Dirichlet distribution, which makes the GP classification task efficiently faster and more scalable. The 7 covariates $\boldsymbol{X}_i \in \mathcal{X}$ in the study are the following: age, years of schooling, african american ethnicity, hispanic ethnicity, marital status, high school diploma. Given the presence of a randomized subsample, we can exploit it to compute unbiased estimates of the quantities of interest and treat them as ground truth. The two quantities of interest in this case are: i) the Average Treatment Effect on the Treated group (ATT), defined as $\text{ATT} = T^{-1} \sum_{i=1}^{T_e} y_i - C^{-1} \sum_{i=1}^{C} y_i$, where $T$ and $C$ are the number of treated and control units in the experimental data; ii) the Policy Risk (Shalit et al., 2017), defined as the average error in allocating the treatment according to the ICE estimates policy rule — namely $\pi(\boldsymbol{x}_i) = 1$ if $\text{ICE} = \mathbb{E}(Y_i|do(A_i = 1), \boldsymbol{x}_i) - \mathbb{E}(Y_i|do(A_i = 0), \boldsymbol{x}_i) > 0$ — or $\mathcal{R}_{\text{pol}} = 1 - \big[\mathbb{E}(Y|do(A_i = 1), \pi(\boldsymbol{x}_i) = 1)p(\pi(\boldsymbol{x}_i) = 1) + \mathbb{E}(Y|do(A_i = 0), \pi(\boldsymbol{x}_i) = 0)p(\pi(\boldsymbol{x}_i) = 0)\big]$. Notice that we cannot measure performance on ICE directly as this is always unobservable in real-world scenarios;

also, we restrict analysis of average causal/treatment effects on the treated group since we are sure that overlap holds there, as all the treated units were part of the randomized experiment subgroup, while the observational subgroup is made only of control units. More details about this experiment can be found in the Appendix C of supplementary materials and in Shalit et al. (2017). We compare the following models: i) GP and CounterGP, as in Alaa & van der Schaar (2017); ii) vanilla PCA plus either GP or CounterGP; iii) vanilla deep AutoEncoder plus either GP or CounterGP; iv) DKL and CounterDKL (ours). Results on performance are gathered in Table 1, in terms of 70%-30% train and test set Mean Absolute Error (MAE) on ATT, Policy Risk $\mathcal{R}_{\text{pol}}$ and average runtime, accompanied by 10-fold cross-validated 95% error intervals. In this example multitasking is induced only over the binary treatment, as we deal with just a single outcome of interest. As the results depict, by operating jointly via a unique loss function, CounterDKL is significantly more efficient than naively applying dimensionality reduction and fitting a multitask GP on a lower dimensional space as two separate steps. It also displays gains over CounterGP, thanks to its deep component that guarantees better computational time (in terms of runtime) and scalability, and is able learn arbitrarily complex functions while imposing weaker smoothness assumptions than standard GP kernels, even on a low-dimensional covariate space example such as the one presented here (7 covariates).

### 5.3 The Infant Health Development Program data

Finally we compare CounterDKL with few other recent methods for causal effects estimation, on a popular simulated experiment utilizing the Infant Health Development Program (IHDP) data, originally found in Hill (2011), and more recently in several contributions on Conditional Average Treatment Effects (CATE) estimation, such as Shalit et al. (2017); Alaa & van der Schaar (2017; 2018); Caron et al. (2022a) among others. The experiment is a semi-simulated setup. It makes use of real-world data from the IHDP study, a randomized clinical trial aimed at improving the health status of premature infants with low weight at birth through pediatric follow-ups and parent support groups, and recreates an observational type of study by removing a non-random portion of treated units, namely those with "non-white mothers". This leaves

| Model | Train RMSE | Test RMSE |
|---|---|---|
| RF | $1.85 \pm 0.13$ | $2.39 \pm 0.17$ |
| X-RF | $3.29 \pm 0.23$ | $3.37 \pm 0.24$ |
| CF | $3.10 \pm 0.21$ | $3.07 \pm 0.20$ |
| BART | $0.97 \pm 0.04$ | $1.44 \pm 0.09$ |
| X-BART | $2.07 \pm 0.14$ | $2.13 \pm 0.14$ |
| BCF | $0.87 \pm 0.05$ | $1.34 \pm 0.10$ |
| **CounterGP** | $\mathbf{0.61 \pm 0.02}$ | $\mathbf{0.70 \pm 0.04}$ |
| **CounterDKL** | $\mathbf{0.62 \pm 0.02}$ | $\mathbf{0.67 \pm 0.04}$ |

Table 2: RMSE of compared models on the semi-simulated IHDP setup, evaluated for CATE estimation on 80%-20% train-test sets.

a total of 139 observations in the treated group and 608 in the control. In addition, the semi-simulated setup uses the real-world binary treatment $A_i \in \{0, 1\}$ and the 25 available covariates $\boldsymbol{X}_i \in \mathcal{X}$, but simulates the two continuous potential outcomes $(Y_1, Y_0) \in \mathbb{R}^2$, as described in the non-linear "Response Surface B" setting in Hill (2011). As anticipated above, the estimand of interest in this case is the Conditional Average Treatment Effects (CATE), defined as $\mathbb{E}[Y_i | do(A_i = 1), \boldsymbol{X}_i = \boldsymbol{x}] - \mathbb{E}[Y_i | do(A_i = 0), \boldsymbol{X}_i = \boldsymbol{x}]$. The models we compare include: i) vanilla Random Forest (**RF**), as a T-Learner (Künzel et al., 2017; Caron et al., 2022a); ii) X-Learner version of Random Forests (**X-RF**) as in (Künzel et al., 2017); iii) Causal Forest (**CF**), or Random Forests as an R-Learner, developed by Wager & Athey (2018); iv) vanilla Bayesian Additive Regression Trees (**BART**), as a T-Learner; v) X-Learner version of BART (**X-BART**); vi) Bayesian Causal Forests (**BCF**) by Hahn et al. (2020); vii) Counterfactual GP (**CounterGP**) as in Alaa & van der Schaar (2018); viii) our Counterfactual DKL (**CounterDKL**) with [100, 100, 2] hidden layers. Results reported in Table 2 refers to 1000 replication of the experiment on 80%-20% train-test split as in Alaa & van der Schaar (2017).

## 6 Conclusions

Throughout this work, we considered the problem of counterfactual effects learning using observational data, which is of interest in domains where exploration of policies is costly (healthcare, socio-economic sciences, etc.). We reviewed the class of counterfactual GP regression models, extending it to adjust to multiple actions and outcomes settings, and discussed how multitask learning over multiple actions helps addressing finite sample selection bias. We then introduced a new class of counterfactual models based on

Deep Kernel Learning, whose main advantages lie in their more flexible function approximation capabilities and better scalability. While counterfactual GPs struggle to scale up with sample size, number of predictors and number of actions/outcomes to coregionalize over, DKL capitalizes on these components by learning lower dimensional representations. We stress that the class of DKL methods proposed can be easily expanded to carry out counterfactual learning in other more complex scenarios such as: i) unobserved confounding where identification is still possible (instrumental variables); ii) dynamic settings such as dynamic treatment regimes or RL; iii) non-standard data like images (as DKL can incorporate any type of deep architecture).

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

# A  Proofs and Discussion

In this first section of supplementary material we provide assumptions, proofs and brief discussion of the two theoretical results in the main paper (Section 2 theorem and Section 3.2 corollary).

## A.1  Backdoor Adjustment

The *backdoor adjustment* theorem (Pearl, 2009) is a well known and understood results in causal inference. Its extension to multi-action and multi-outcome settings is trivial. We briefly reformulate the necessary assumptions as follows.

**Assumption A.1** (Backdoor Criterion). Using the causal graph terminology, we denote by $pa(V_j)$ the parents of a specific random variable $V_j$ and by $de(V_j)$ its descendants. We assume that $pa(A, \boldsymbol{Y}) \subseteq \boldsymbol{X}$, $de(A, \boldsymbol{Y}) \not\subset \boldsymbol{X}$. Namely $\boldsymbol{X}$ contains (but does not necessarily coincide with) all the common parent variables of $A \in \mathcal{A}$ and $\boldsymbol{Y} = \{Y_m\}_{m=1}^M \in \mathcal{Y}$, for all $m \in \{1, ..., M\}$, and does not contain any common descendant of them.

**Assumption A.2** (Overlap). Defining the propensity score as $\pi_a(\boldsymbol{x}_i) = p(A_i = a | \boldsymbol{X}_i = \boldsymbol{x}_i)$, we require that $\pi_a(\boldsymbol{x}_i) \in (\alpha, 1 - \alpha)$ for all $i \in \{1, ..., N\}$, where $\alpha \in (0, \frac{1}{2})$. The scalar $\alpha$ represents how tight we require the overlap between units in each arm in terms of the covariates to be.

**Proof of Backdoor Adjustment Theorem**  Under Assumption 1.1 and 1.2, we are able to identify the effect $A \to \boldsymbol{Y} = \{Y_m\}_{m=1}^M$, in the form of the joint interventional distribution $p(\boldsymbol{Y} | do(A = a))$, as:

$$
\begin{aligned}
p(\boldsymbol{Y} \mid do(A = a)) &= \int_{\mathcal{X}} p(\boldsymbol{Y} \mid do(A = a), \boldsymbol{x}) p(\boldsymbol{x} \mid do(A = a)) d\boldsymbol{x} \\
&= \int_{\mathcal{X}} p(\boldsymbol{Y} \mid a, \boldsymbol{x}) p(\boldsymbol{x} \mid do(A = a)) \, d\boldsymbol{x} \\
&= \int_{\mathcal{X}} p(\boldsymbol{Y} \mid a, \boldsymbol{x}) p(\boldsymbol{x}) \, d\boldsymbol{x},
\end{aligned}
$$

The above derivation refers to the marginal interventional distribution, with respect to $\boldsymbol{X}$. Often we are interest in the conditional interventional distribution $p(\boldsymbol{Y} | do(A = a), \boldsymbol{X})$ instead (e.g., conditional on patient's characteristics), such as when estimating CATE: $\tau(\boldsymbol{x}) = f_1(\boldsymbol{x}) - f_0(\boldsymbol{x})$. The only additional requirement compared to the original version of backdoor adjustment (Pearl, 2009) is that Assumption 1.1 holds for all the collection of outcomes $\boldsymbol{Y}$.

## A.2  Corollary 3.1

Corollary 3.1 is a trivial extension of the result on CATE optimal minimax rate derived in Theorem 1 by Alaa & van der Schaar (2018) to discrete multi-action set domains, where specifically $\{0, 1\} \subset \mathcal{A}$. Thus the proof follows straightforwardly from Alaa & van der Schaar (2018). The main difference is the following.

**Proof of Corollary 3.1**  Optimal minimax rate in Alaa & van der Schaar (2018) define the hardness of CATE estimation between binary action $a, b \in \mathcal{A}$: $\tau_{a,b}(\boldsymbol{x}_i) = f_a(\boldsymbol{x}_i) - f_b(\boldsymbol{x}_i)$. If we have multi-actions space, it means that, given the whole sample size is $N$, each pair (strictly more than one pair) of discrete actions $a, b$ defines a subsample (and thus a subpopulation) of $n_{a,b} < N$ units. This implies that the hardness of approximating a function in the Hölder ball class and of performing variable selection is proportional to the smaller subsample $n_{a,b}$, not $N$, which makes the CATE estimation problem harder the smaller $n_{a,b}$.

This is likely to happens when multi-action scenarios feature infrequently explored action arms. Thus, technically speaking, result in Theorem 1 in Alaa & van der Schaar (2018) is a special case of Corollary 3.1 where $\mathcal{A} = \{0, 1\}$.

# B  Data Generating Processes

We hereby describe the causal data generating processes in the simulated examples of the paper (Section 3.2 and Section 5.1).

## B.1  Section 3.2 one covariate example

For the simple one-covariate example in Section 3.2 (Figure 2), where we discuss the benefits of multitask counterfactual learning, we generated $N = 300$ data points from one, uniformly distributed covariate, $X_i \sim \text{Uniform}(-3, 3)$. Then we generated a binary action variable $A_i \sim \text{Bernoulli}\big(p(A_i = 1|x_i)\big)$, where $p(A_i = 1|x_i) = \Phi\big(0.2 + X_i\big)$ and $\Phi(\cdot)$ is the standard normal cdf. Finally, the two counterfactual outcome surfaces were generated as $f_0(x_i) = 2 + 0.3 \exp X_i$ and $f_1(x_i) = 3 + f_0(x_i)$, with the final outcome being $Y = f_0(x_i) + \tau(x_i)A_i + \varepsilon_i$ where $\tau(x_i) = f_1(x_i) - f_0(x_i)$ is the CATE function and $\varepsilon_i \sim \mathcal{N}(0, 0.75^2)$.

## B.2  Section 5.1 experiment

The causal data generating process for the simulated experiment of Section 5.1 is described as follows. The $P$ covariates are generated from a uniform distribution $X_{i,j} \sim \text{Unif}(-3, 3)$ for $j \in \{1, ..., P\}$ and $i \in \{1, ..., N\}$. The action allocation policy is simulated according to a multinomial distribution where the probabilities of being assigned to action $A_i = a$ are generated as a softmax function of the covariates $p(A_i = a|\boldsymbol{X}_i = \boldsymbol{x}_i) = \exp\{X_i\boldsymbol{\beta}_a\} / \sum_{a \in \mathcal{A}} \exp\{X_i\boldsymbol{\beta}_a\}$, where $\boldsymbol{\beta}_a$ is an action-specific $P$-dimensional sparse vector of action-specific coefficients defined as follow:

$$\boldsymbol{\beta}_1 = \begin{bmatrix} -1 & -0.8 & -0.1 & -0.1 & 0 & ... & 0 \end{bmatrix},$$
$$\boldsymbol{\beta}_2 = \begin{bmatrix} 0 & 0 & 1 & 0.8 & 0.2 & 0 & ... & 0 \end{bmatrix},$$
$$\boldsymbol{\beta}_3 = \begin{bmatrix} 1.5 & -0.8 & -0.1 & -0.1 & 0 & ... & 0 \end{bmatrix},$$
$$\boldsymbol{\beta}_4 = \begin{bmatrix} -1 & -0.8 & -0.1 & -0.1 & 0 & ... & 0 \end{bmatrix}.$$

Thus $A_i$ is drawn from a multinomial with vector probabilities parameter $\boldsymbol{p}(A_i = a|\boldsymbol{X}_i = \boldsymbol{x}_i)$. The $M = 2$ action-specific correlated counterfactual outcomes $\boldsymbol{Y}_i \mid do(A_i = a)$ instead are generated as

$$\boldsymbol{Y}_i \mid do(A_i = a) = \boldsymbol{f_Y}_a(\boldsymbol{X}_i) + \boldsymbol{\varepsilon}_i, \quad \boldsymbol{\varepsilon}_i \sim \mathcal{N}(\boldsymbol{0}, \Sigma_{\varepsilon_i}), \quad \text{where:}$$

$$f_{Y11} = 3 + 0.4X_0X_1 - 0.3X_2^2 + 0.2\exp(X_3) + 0.6\sin(X_4)$$
$$f_{Y12} = -1 + f_{Y11} + 0.1X_5$$
$$f_{Y13} = 1 + f_{Y11} + 0.3X_5$$
$$f_{Y14} = 0.5 + f_{Y11} + 0.5X_6$$

$$f_{Y21} = 1 + 0.2X_0X_1 - 0.2X_2^2 + 0.1\exp(X_3)$$
$$f_{Y22} = -2 + f_{Y21} + 0.2X_5$$
$$f_{Y23} = 2 + f_{Y21} + 0.4X_5$$
$$f_{Y24} = 1 + f_{Y21} + 0.5X_6$$

and where $\text{diag}(\Sigma_\varepsilon) = [\sigma_1, ..., \sigma_4]$, with $\sigma_1 = ... = \sigma_4 = 0.5$, and off-diagonal elements are 0. Finally, we briefly describe the main specifications of the methods compared. The GP models (GP, CounterGP and MOGP) all employed a RBF base kernel, while the DKL models employed a three [50, 50, 2] hidden layers feedforward neural network before the GP $\infty$-layer, which itself employs a RBF base kernel. The multitask and multioutput models (both GPs and DKLs) all make use of the Intrinsic Coregionalization Model (ICM), such that $K(\boldsymbol{x}_i, \boldsymbol{x}_i') = B_Y \otimes B_A \otimes K_q(\boldsymbol{x}_i, \boldsymbol{x}_i')$. All model were optimized through the Adam solver. More details and fully reproducible code on this experiment can be found in the Github repository: https://github.com/albicaron/CounterDKL.

| Data | $N$ | $P$ | # actions |
|------|-----|-----|-----------|
| indian | 573 | 10 | 2 |
| heart | 270 | 13 | 2 |
| yeast | 1484 | 8 | 10 |
| contracept | 1473 | 9 | 3 |

Table 3: UCI datasets characteristics.

| | GP | CounterGP | DKL | CounterDKL |
|--------|------|-----------|------|------------|
| indian | 0.390 | 0.392 | 0.376 | **0.347** |
| heart | 2.553 | 1.076 | 0.433 | **0.410** |
| yeast | 0.534 | 0.657 | 1.3144 | **0.081** |
| contracep | 0.339 | **0.003** | 0.008 | 0.007 |

Table 4: OPE absolute regret on UCI datasets. Bold denotes best performance.

## C The Job Training Data

The Job Training data (LaLonde, 1986) are a popular case study in the causal inference literature. They comprise a portion of data pertaining to a randomized experiment and a portion of observational data, with the randomized experiment featuring 297 treated and 425 control units, while the observational data being of 2490 control units only. Given the randomized subsample of the data, we can obtain an unbiased estimate (computed on the randomized units only) for the Average Treatment Effect on the Treated group (ATT) as $\text{ATT} = T^{-1} \sum_{i=1}^{T_e} y_i - C^{-1} \sum_{i=1}^{C} y_i$, where $T$ and $C$ are the number of treated and control units in the experimental data, and treat this as the ground truth for estimating performance of the methods; and also for the policy risk measure $\mathcal{R}_{\text{pol}} = 1 - \left[ \mathbb{E}\big(Y|do(A_i = 1), \pi(\boldsymbol{x}_i) = 1\big)p(\pi(\boldsymbol{x}_i) = 1) + \mathbb{E}\big(Y|do(A_i = 0), \pi(\boldsymbol{x}_i) = 0\big)p(\pi(\boldsymbol{x}_i) = 0)\right]$.

A brief overview on the specifications of the models employed follows. All GPs employ RBF base kernel (also DKL's specifications in the last hidden layer). DKL and CounterDKL deep NN structure features three [10, 5, 2] hidden layers. The AutoEncoder deep structure employed for the "AutoEnc + GP" and "AutoEnc + CounterGP" models similarly learns a 2-dimensional encoded lower-dimensional representation, where the encoder has two [10, 5] hidden layers before the 2-dim representation and the decoder has [5, 10] hidden layers before the reconstruction loss.

## D Marginal Likelihood Maximization in Multioutput Deep Kernels

In the multitask deep kernel learning class of models, the parameter space $\Theta = (W, \phi, B)$ is made of the deep neural network's weights $W$, the base kernel's hyperparameters $\phi$ (variance, lengthscales, etc.) and the coregionalization matrix $B$ entries. These parameters are learnt jointly by maximizing the log-marginal likelihood $\mathcal{L}$ at the end of the GP layer. Using the chain rule, the derivatives are:

$$\frac{\partial \mathcal{L}}{\partial W} = \frac{\partial \mathcal{L}}{\partial K_\phi} \frac{\partial K_\phi}{\partial g(\boldsymbol{x}, W)} \frac{\partial g(\boldsymbol{x}, W)}{\partial W}$$

$$\frac{\partial \mathcal{L}}{\partial \phi} = \frac{\partial \mathcal{L}}{\partial K_\phi} \frac{\partial K_\phi}{\partial \phi}$$

$$\frac{\partial \mathcal{L}}{\partial B} = \frac{\partial \mathcal{L}}{\partial K} \frac{\partial K}{\partial B}$$

where $g(\boldsymbol{x}, W)$ is the function mapping the inputs to the lower representation space parametrized by $W$, $K_\phi$ is the base kernel and $K(\cdot) = B \otimes K_\phi(\cdot)$ is the coregionalized kernel.

## E Additional Simulated Experiments

Finally, we describe and present results on a few additional simulated examples that we conducted to assess CounterDKL performance compared to some other specifications seen in Section 5.2, on datasets with varying sample size, predictor space and action space dimensions. In particular, following Dudík et al. (2011) and Farajtabar et al. (2018), we make use of some of the popular datasets for classification in the open-source UCI Machine Learning Repository (https://archive.ics.uci.edu/ml/index.php), by transforming the

classification task in a causal Off-Policy Evaluation task in the following way. Each dataset is equipped with a pair of covariates $\boldsymbol{X}_i$ and classification labels $L_i$. We view the classification labels $L_i$ as our discrete actions $L_i = A_i$, and consequently generate the action-specific outcome $Y_{a_i}$ as function of the covariates as follows:

$$Y_{a_i} = \exp\{\boldsymbol{X_i}\boldsymbol{\beta_a}\} + \varepsilon_i, \quad \text{where} \quad \mathcal{N}(0, 0.5)$$

and $\boldsymbol{\beta_a}$ is a $P$-dimensional vector of action-specific coefficients, where entries are $\{0.4, 0.2, 0.0\}$ sampled from a Multinomial$(0.6, 0.25, 0.15)$, with replacement. The datasets utilized are summarized in Table 3 in terms of sample size $N$, number of covariates $P$ and number of actions. We compare GP, CounterGP, DKL and CounterDKL models on an Off-Policy Evaluation task, where we evaluate the uniformly at random generated policy, via the absoulte regret or risk measure, defined as $\mathbb{E}\big[\, |\, \mathcal{V}(\pi_e) - \hat{\mathcal{V}}(\pi_e)\,|\,\big]$. All models employ a RBF base kernel, either directly on the inputs or on the lower dimensional layer. Results averaged over $B = 20$ replications of the experiments for each dataset are gathered in Table 4.

