# OpenReview forum: "Counterfactual Learning with Multioutput Deep Kernels"
_TMLR — Accepted by TMLR_

### Review · Reviewer_65W5 · 2022-07-21

**Summary Of Contributions:**

This paper addresses the problem of performing causal inference in observational data, in settings where we have a large number of covariates, multiple discrete actions, and multiple correlated outcomes of interest. The authors propose CounterDKL (i.e., counterfactual multi-task Deep Kernel Learning method), which first learns a lower dimensional representation space of the covariates, and then places a multi-task Gaussian process (GP) on top of the learned representation. The simulated experiments demonstrate the benefits of the proposed method.

**Requested Changes:**

- Discuss explicitly and clearly, what are the contributions made on top of (Alaa and van der Schaar, 2017, 2018), besides employing the Deep Kernel Learning component.
- Address the mentioned weaknesses in the previous box.


**Strengths And Weaknesses:**

Strengths
- Very good Introduction section.
- The paper is written clearly and easy to read.
- Empirical experiments are thorough and show the merits of the proposed method.

Weaknesses
- Contributions of the paper are stated under the Related Works section. The convention is that it’s stated at the end of the Introduction section.
- Methods proposed in (Shalit et al, 2017), (Yao et al, 2018), etc. are not non-parametric. They are parametric methods, parameterized by neural networks.
- In description of the overlap assumption, it is stated that $p(A_i=a | X_i=x) \in (\alpha, 1-\alpha)$ where $\alpha \in (0, 0.5]$. Why "$0.5]$"? Shouldn’t this be "$1)$"? We only need non-zero probability on the entire support.
- In Section 3.1, it is not clear what $Q$ and $R_q$ are, and how they are related. It is also not clear what $K_q$ is, and how it is related to $K$.
- There is little elaboration on the training process: What are the learnable parameters? What is the objective function (if any)? Is the training end-to-end?

---

> ### Author Response · Authors · 2022-08-31
> **Response to Reviewer 65W5's comments**
>
> We thank the reviewer for the time spent revising our manuscript and for the nice comments about the strengths of the paper. We proceed by discussing how the requested changes have been addressed.
>
> In order to more easily link back manuscript changes to the reviewer-specific comments when we will upload the new version, we have used text with different colors in the revised version of the manuscript. Changes related to Reviewer 65W5's comments are in **blue** text.
>
>
> # Response to comments:
>
> ---
>
> ## Weaknesses
>
> - Contributions of the paper are stated under the Related Works section. The convention is that it’s stated at the end of the Introduction section.
>
> **True, we have moved the contributions paragraph to the Introduction section in the new version of the manuscript, thank you.**
>
> - Methods proposed in (Shalit et al, 2017), (Yao et al, 2018), etc. are not non-parametric. They are parametric methods, parameterized by neural networks.
>
> **This is true. We have now amended this typo by changing the wording in the paragraph to the more general “non-linear regression models” terminology instead of “non-parametric”, thank you.**
>
> - In description of the overlap assumption, it is stated that p(Ai=a|Xi=x)∈(α,1−α) where α∈(0,0.5]. Why "0.5]"? Shouldn’t this be "1)"? We only need non-zero probability on the entire support.
>
> **Indeed, it is true that the most relaxed version of the overlap assumption only requires that $0 < p(A_i = a | X_i = x) < 1$ for each action arm. We have now amended this statement in the new version of the paper.**
>
> - In Section 3.1, it is not clear what Q and Rq are, and how they are related. It is also not clear what Kq is, and how it is related to K.
>
> **We have now substantially re-written the paragraphs at the end of Section 3, Section 3.1 and partly Section 4 in the new version of the paper, in the attempt to better clarify the (quite heavy) notation of separable multi-output kernels and coregionalization matrix.**
>
> - There is little elaboration on the training process: What are the learnable parameters? What is the objective function (if any)? Is the training end-to-end?
>
> **We reasonably assume this refers to the training of Counter DKL models (Section 4). We have now added more explanation about the learnable parameters in the revised version of Section 4 and more details on the training loss function in the Appendix D.**
>
> ---
>
> ## Requested Changes
>
> - Discuss explicitly and clearly, what are the contributions made on top of (Alaa and van der Schaar, 2017, 2018), besides employing the Deep Kernel Learning component.
>
> **We have added clarifications about how we have extended Alaa and van der Schaar (2017, 2018) work in the introductory section and in the revised version of Section 3.1 and Section 4 in more distinct/separate way.**
>
> - Address the mentioned weaknesses in the previous box.
>
> **See weaknesses section above.**
>
> ---
>
> &nbsp;
>
> Thank you!

---

### Review · Reviewer_6xmF · 2022-07-27

**Summary Of Contributions:**

This paper considers the problem of learning multiple counterfactual functions from observational data for causal inference. In particular, the manuscript focuses on the high-dimensional scenario with multiple actions and correlated outcomes. This becomes particularly convenient for multi-task Gaussian processes (GPs), that are scaled up using *deep kernels*. In the paper, the methodology for introducing multioutput deep kernels in the counterfactual learning problem is presented, followed by results showing the benefits of modelling the correlations among such functions.

**Broader Impact Concerns:**

I have no concerns on this

**Requested Changes:**

Additionally, I want to remark that authors are aware of the limitations that introducing DKL (Wilson et al.  2016) might have for tuning parameters, which are deeper discussed in Ober et al. (2021). Here, I would appreciate if authors could address/answer some of the questions and points of change that wrote down while reading the manuscript.

**Questions**

 - **Q1** --  In the Introduction, it is written *"exploration of policies in the real-world is usually very costly and potentially harmful"*. Why harmful? Could you elaborate on a particular example?
 - **Q2** -- How restricted is the model to accept discrete actions -- i.e. binary. Could the authors indicate what would be the main changes for considering other continuous actions or similar?
 - **Q3** -- Why first equation in Section 3.3. is not like Eq. (2)? where did A_i go?
 - **Q4** -- Is CounterGP one of the models considered in Alaa & van der Schaar (2017,2018)?
 - **Q5** -- Can correlations vanish due the representation learning task (i) in the 2nd paragraph of the intro) of CounterDKL?

**Changes**

- **C1** -- I would love to see more details on Section 3.3., particularly on how both correlation matrices B_y and B_A emerge from the formulation. That would be fantastic for the paper.
- **C2** -- Experiments are nice! but somehow baselines are a bit disconnected from the SOTA to me. Could authors put which model corresponds to what paper?
- **C3** -- Experiment in Section 5.2. is important as it is the application one. A bit more of explanation on the particular properties of the data and the target task would help.
- **C4** -- The paper claims a lot the DKL for high-dimensional scenarios, but later experiments are somehow on data with ~ 10 covariates. Would it be possible to show the performance on a larger dataset or task? At least to perceive the type of scalability of the counterfactual model.
- **C5** -- I also think it would be important to compare the method with one of the experiments in Alaa & Schaar 2017, 2018 if it is possible.
- **C6** -- Perhaps, explaining better or introducing the *policy-based* metrics in Section 5.1 would help.

**Minors**

- $\epsilon$ corresponding to $\mathcal{U}$ exogeneous variables is a bit confusing. I see that it is because of the latent functions $u$ in the MOGP. But perhaps there is a better choice...
- *guarantees* just before Section 2.

**Strengths And Weaknesses:**

First of all, I remark that I liked the manuscript and enjoyed while reading it. The motivation, presentation, notation and development of the main ideas and methodology is very clear and thorough to me. Overall, the quality of the paper is high in its current state, with perhaps a few details or points for improvement.

**Strengths**

- The paper adequately identifies the benefits of introducing multi-output GPs in the counterfactual learning problem for exploiting the existing correlations.
- Clarity is excellent in the manuscript, even for readers non-familiar with the specific problem of counterfactual learning.
- The work is novel, as it extends *causal multi-task GPs* (Alaa & van der Schaar NIPS 2017) for accepting a stacked corregionalization model, and also introduces deep kernel learning for treating with high-dimensional covariates.
- Experiments show empirical results of the performance and how the multi-output flavour helps when considering correlations between counterfactual functions.

**Weaknesses**

- Presentation of tasks based on *policies* could be better/longer explained for a better understanding, as it does not seem completelty clear to me.
- Experimental results could be slightly extended to match one of the particular results included in (Alaa & van der Schaar 2017,2018). Since this is the main reference in the SOTA for the work.
- Theoretical results in Section 3.2 are of interest for the manuscript, but perhaps *Corollary 3.1* could be better explained or at least explain somewhere how Eq. (5) is obtained.

---

> ### Author Response · Authors · 2022-09-05
> **Response to Reviewer 6xmF's comments (Part 1)**
>
> We thank Reviewer 6xmF for taking the time to revise out manuscript. We particularly appreciated comments about the quality, readability and clarity of exposition.
>
> As in the case of the first reviewer, we have used text with different colors to help linking make changes and reviewers’ comments. Changes related to Reviewer 6xmF’s comments are in **green** text.
>
> # Response to Comments (Part 1)
>
> ---
>
> ## Weaknesses
>
> •	Presentation of tasks based on policies could be better/longer explained for a better understanding, as it does not seem completelty clear to me.
>
> **Indeed, we have added further clarification to each of the three policy tasks in the corresponding Section 5.1 in the revised version of the manuscript.**
>
> •	Experimental results could be slightly extended to match one of the particular results included in (Alaa & van der Schaar 2017,2018). Since this is the main reference in the SOTA for the work.
>
> **We have added a new simulated example that matches one of Van Der Schaar's (2017) involving the IHDP dataset in the new Section 5.3, to compare CounterDKL with few other SOTA methods in addition to Van Der Schaar's (2017) CounterGP. The main difference to Van Der Schaar's (2017) is that we evaluate methods on the estimation of Conditional Average Treatment Effects on the Treated (CATT) group portion only, as Conditional Average Treatment Effects on the Controls is not identifiable, due to lack of overlap - i.e. following Hill (2011), VdS (2017, 2018), Caron et al. (2022), we throw away a portion of treated units with “non-white mothers” value in the covariates to recreate an observational study, so we are no longer able to find a comparison unit in the treated, for “non-white” mothers present in the control (violation of the overlap assumption, see Caron et al., 2022 for reference).**
>
> **We are open for discussion/suggestions as to whether include this simulated example here or another simulated example requested by Reviewer 3 (that is featuring in the Appendix Section E at the moment) in the main body of the paper. Or also both, page constraints permitting**
>
> •	Theoretical results in Section 3.2 are of interest for the manuscript, but perhaps Corollary 3.1 could be better explained or at least explain somewhere how Eq. (5) is obtained.
>
> **We have added slightly more information/intuition for Corollary 3.1 in the new version, but more details about derivation can be found in the Appendix section A.2, and furthermore in Alaa & Van Der Schaar (2018).**

---

> ### Author Response · Authors · 2022-09-05
> **Response to Reviewer 6xmF's comments (Part 2)**
>
> # Response to comments (Part 2)
>
> ---
>
> ## Requested Changes
>
> ### Questions
>
> •	Q1 -- In the Introduction, it is written "exploration of policies in the real-world is usually very costly and potentially harmful". Why harmful? Could you elaborate on a particular example?
>
> **Yes, we have provided a simple example in the related paragraph.**
>
> •	Q2 -- How restricted is the model to accept discrete actions -- i.e. binary. Could the authors indicate what would be the main changes for considering other continuous actions or similar?
>
> **The settings with continuous action space, concerned with the estimation of a dose-response curve, are a non-trivial extension, as these imply having uncountably many interventional distributions $p(Y_i | do(A_i = a), X_i)$ for $a \subset \mathbb{R}$. In this setting the multitask paradigm for causal inference is not suitable as this requires the sample to be splitable in D different sub-sample (as many as the correlation off-diagonal entries of the D x D coregionalization matrix) defined by the D action arms. We have added a footnote about this, also citing some very recent works on dose-response curve estimation.**
>
> •	Q3 -- Why first equation in Section 3.3. is not like Eq. (2)? where did A_i go?
>
> **Yes, this is typo. The general point is that we assume in the SCM that the outcome is some function $Y_i = f_Y (X_i, A_i, \varepsilon_i)$, but in practice we tackle estimation in the discrete action case by stratifying on values of $A_i$, so we effectively estimate a set of functionals $f_{Y_a} (\cdot)$ indexed by $a$. So in Section 3.3, eq. 7, $f_{Y}$ should also be indexed by $a$ as $f_{Y_a} (\cdot)$. We corrected this typo also in the recurrences of Section 4, thank you.**
>
> •	Q4 -- Is CounterGP one of the models considered in Alaa & van der Schaar (2017,2018)?
>
> **Yes, it is, with the only difference being the training algorithm and the python package utilized. We have added clarification. Please response to C2 below for full details.**
>
> •	Q5 -- Can correlations vanish due the representation learning task (i) in the 2nd paragraph of the intro) of CounterDKL?
>
> **If this refers to correlation among actions (i.e. correlation in the way different action arms affect the outcome), we don’t think it is the case. The reason is that different actions displaying correlation is due to selection bias and the fact that there is a “nuisance” baseline effect of the covariates on the outcome, sometimes referred to as \emph{prognostic effect} (the effect of covariates on the outcome regardless of which treatment/action is performed – e.g. the effect of age on cholesterol level, regardless of whether patient is prescribed statin), that is shared across every action arm. So learning lower dimensional representations of X does not really affect their “nuisance” \emph{prognostic effect}.**

---

> ### Author Response · Authors · 2022-09-05
> **Response to Reviewer 6xmF's comments (Part 3)**
>
> # Response to Comments (Part 3)
>
> ---
>
> ### Changes
>
> •	C1 -- I would love to see more details on Section 3.3., particularly on how both correlation matrices B_y and B_A emerge from the formulation. That would be fantastic for the paper.
>
> **In response also to other reviewers’ comment, we have now substantially re-written Section 3.1 to correct typos in the notation and better clarify how the actions’ coregionalization matrix $B_A$ is derived in the multitask framework. We have also added a few lines on how $B_y$ is formulated in Section 3.3 (in a very similar fashion as $B_A$ indeed). Hope this helps!**
>
> •	C2 -- Experiments are nice! but somehow baselines are a bit disconnected from the SOTA to me. Could authors put which model corresponds to what paper?
>
> **True, we have now added reference in Section 5.1 saying that CounterGP corresponds to the work of Alaa and Van Der Schaar (2017, 2018), as this is the only SOTA reference for the simulations (with the only difference from Van Der Schaar (2017, 2018) that we train using GPytorch package using Adam solver).**
>
> •	C3 -- Experiment in Section 5.2. is important as it is the application one. A bit more of explanation on the particular properties of the data and the target task would help.
>
> **Main details of the Lalonde study previously featured in the Appendix Section C. We have now moved some of the salient info directly in Section 5.2 (reasonably in compatibility with pages constraints)**
>
> •	C4 -- The paper claims a lot the DKL for high-dimensional scenarios, but later experiments are somehow on data with ~ 10 covariates. Would it be possible to show the performance on a larger dataset or task? At least to perceive the type of scalability of the counterfactual model.
>
> **In response to one of the other comments we have added the IHDP simulated example of Van Der Schaar, where covariates included are 25 (in Section 5.3).**
>
> •	C5 -- I also think it would be important to compare the method with one of the experiments in Alaa & Schaar 2017, 2018 if it is possible.
>
> **As said above, we have added comparison to the example in Van Der Schaar (2017) on the IHDP semi-simulated dataset in the new Section 5.3.**
>
> •	C6 -- Perhaps, explaining better or introducing the policy-based metrics in Section 5.1 would help.
>
> **Done, see one of the other comments above, thank you!**
>
> ### Minors
>
> •	ϵ corresponding to U exogeneous variables is a bit confusing. I see that it is because of the latent functions u in the MOGP. But perhaps there is a better choice...
>
> **We have replaced $\mathcal{U}$ with $\mathcal{E}$ to denote exogenous variables in the definition of the SCM, to be consistent with the rest and avoid confusion.**
>
> •	guarantees just before Section 2.
>
> **Amended, thank you.**
>
> ---
>
> ### Thank you!

---

### Review · Reviewer_9sDy · 2022-08-26

**Summary Of Contributions:**

The paper proposes a method for estimating counterfactual outcome surfaces using Gaussian processes that are parameterized using the DKL framework. The Gaussian processes are linked using a Kronecker structured covariance kernel that allows information sharing across counterfactuals that correspond to different treatments levels, and across potentially multi-dimensional outcomes. The authors compare the overall Multi-Output Counter DKL approach to ablated baselines in a simulation study, and compare the single outcome approach to previous GP-based approaches on the well-established LaLonde job training data.

**Broader Impact Concerns:**

No concerns.

**Requested Changes:**

# Highlighting the main contributions

The main technical contributions are (1) introducing a new covariance structure that scales up the Kronecker product corresponding to the number of counterfactual outcome surfaces, and to adds a new Kronecker product corresponding to the outcome dimension, and (2) using the DKL structure to reduce the dimension of the GP on the covariate space, perhaps making the expanded covariance structure more tractable. I think focusing more of the paper on precisely describing the scaling challenges that this introduces and how DKL addresses them would be more appropriate. As of now, these technical contributions are difficult to pull out, where more than half of the paper is dedicated to restating known results.

Slimming down the background material add adding more empirical exploration could be a useful way to reframe the paper.

Some examples of unnecessary exposition:
 - Figure 1(b) and 1(c) are unnecessary. The working assumption is that $\vX$ satisfies the backdoor criterion, so discussing other cases is not relevant to this paper.
 - Theorem 2.2 is not necessary (it is also wrong, see below). Simply quoting the backdoor adjustment formula would be sufficient.
 - Corollary 3.1 does not seem necessary. It is an asymptotic result, not finite sample, and the main point that it becomes hard to estimate an outcome surface when there are sparsely sampled input regions for a particular counterfactual does not require a formal statement. In fact, the target of estimation in this paper is not causal effects, but the intervention distributions themselves, so the primary content of this corollary (which relates the difficulty of estimating a contrast between surfaces to the difficulty of estimating each individual surface) is not relevant to the problem being solved. In terms of the challenges that result from many levels of A, simply stating that the sample size gets sharded across these actions, often very unevenly, seems to make this point effectively.

# Exploring bias-variance tradeoffs

The authors note that the primary benefit of the multi-task GP framework is sharing information across outcome surfaces (either between counterfactuals or between outcome dimensions). However, this information sharing introduces bias. Right now the paper does not do a good job of noting the kind of risk that this bias can introduce, and the settings under which the bias would be beneficial. For example, in Figure 2, the red surface could have just as easily continued horizontally, in which case the multi-task GP estimate would be very far off.

One additional risk here is that the residual correlation between counterfactual outcomes---say, Y_a1, Y_a2---after conditioning on X is not identifiable, even with infinite data, but the proposed approach (and CounterGP) specify a correlation between them. If this specification introduces bias in the estimated intervened expectation surfaces, it would not fade away with larger sample size. This would also certainly affect the joint uncertainty bands associated with Y_a1 and Y_a2.

It would be useful to specify some simulations where the multi-task GP helps and some where it hurts (at least in finite samples), and see how the benefit fades with larger sample size.

# More detailed empirical exploration

The empirical exploration in this paper could be much stronger. Evaluation of Bayesian uncertainty from the model, and evaluations on a wider range of simulations that showcase different aspects and tradeoffs of model performance would be a good start. Also, specifying a problem where the multi-outcome mutli-task GP clearly helps would be useful. Finally, demonstrating the method on a real dataset that has the multi-action-level multi-outcome structure would provide far stronger motivation. The authors claim that this is a common setting, so it should be doable to find such a dataset.

# Misstatements in Section 2 on Causal Identification Conditions

A DAG is not equivalent to an SCM. The DAG does not specify the F or p(u) part of the tuple. A better statement would be that an SCM implies a DAG.

The statement of Theorem 2.2 is incorrect. The backdoor adjustment formula says that p(\vY | do(A=a), \vX) = p(\vY | A=a, \vX). From this P(\vY | do(A=a)) is recovered by E[p(\vY | A=a, X)], where this expectation is taken over the marginal distribution of X, p(X). Note that P(\vY | A=a) = E[p(\VY | A=a, X) | A=a], which is very different, even if the backdoor criterion is satisfied. The last equality in the proof is wrong (it would only hold if p(x | A=a) = p(x)).

Theorem 2.2 is also unnecessary. The statement of the do calculus puts no restrictions on the dimensionality of the variables in the graph, so this statement for multivariate Y is already implied by well-established results.

“Provided that covariates X (or a subset of them) satisfy the backdoor criterion”. Note the famous M-bias or M-structure example in which a subset of X satisfy the backdoor criterion, but adjusting for all of X does not. I.e., satisfaction of the backdoor criterion of a subset of variables does not imply that the superset satisfies the backdoor criterion.

# Other Comments

The quantity Y_i|do(A_i=a_i) is usually written as a counterfactual Y_{a} or a potential outcome Y(a). do notation is reserved for distributions (hence its use as a conditioning variable).

The LHS of the definition equation right before (4) does not really parse. Should we be conditioning on f_Y, or on m(.) and K(.,.)? If conditioning on f_Y, it seems like f_Y(x*_j) is just a constant. Meanwhile, eq (4) is written entirely in terms of the hyperparameters m and K.

The indexing of the coregionalization matrix is extremely difficult to parse. In the end, since Q is fixed to be 1, it seems like it would make sense to just present the intrinsic coregionalization matrix here, and note that it can be generalized to the LMC.

The LHS’s of the math environment on page 5 don’t seem to be consistent. In the second equation, should the arguments of k(.,.) be f_d(\vx_i) or should they just be \vx_i? It seems like the latter, given than the RHS is written in terms of \vx_i.

**Strengths And Weaknesses:**

Strengths:
 - Combines three idea from the literature (CounterGP, DKL, and more general multi-task GP's) in a novel way to attack causal inference problems that are high-dimensional along multiple axes (many treatment levels, many outcomes, many covariates).
 - Experiment has a nice ablation design.

Weaknesses:
 - In the presentation, too much time is spent reviewing background material. The key idea of using DKL is not introduced until page 7. Some of the background material is not particularly necessary or relevant (for example, there is no need to restate the backdoor adjustment formula as a theorem, and the re-statement of the minimax bound in Corollary 3.1 does not really speak to finite-sample performance---see requested changes).
 - Relatedly, the main technical contributions are difficult to parse out from the exposition. See requested changes for proposed focus.

 - The tradeoffs introduced by the modeling assumptions built into the framework are not well-described. There is a bias-variance tradeoff that comes from sharing information between surfaces (either between outcomes, or between counterfactuals). This is particularly relevant for sharing information between counterfactuals, because the correlation between counterfactuals is fundamentally unidentifiable from the data.
 - Relatedly, the experiments could be more thorough in exploring tradeoffs, particularly whether bias introduced by the GP prior really does fade with sample size.

 - Despite using a fully Bayesian framework and plotting credible bands in some of the demo figures, there is no evaluation of Bayesian uncertainty from the proposed method.
 - The experiments are not particularly compelling for the multi-outcome approach. It would be useful to at least specify a simulation where one can see the lift from the multi-outcome approach.
 - The authors also don't give an example of a real dataset that matches the multi-valued treatment multidimensional outcome setting that they built their method for.

 - There are several misstatements about causal inference with backdoor adjustment.

---

> ### Author Response · Authors · 2022-09-09
> **Response to Reviewer 9sDy's comments (Part 1)**
>
> # Response to Reviewer 9sDy's comments (Part 1)
>
> We thank the reviewer for the time spent carefully revising our manuscript. We discuss below how the requested changes have been addressed.
>
> In order to link back manuscript changes more easily to the reviewer-specific comments when we upload the new version, we have used text with different colours in the revised version of the manuscript. Changes related to Reviewer 9sDy's comments are in **red** text.
>
> &nbsp;
>
> # Requested Changes:
>
> &nbsp;
>
> ## Highlighting the main contributions
>
> The main technical contributions are (1) introducing a new covariance structure that scales up the Kronecker product corresponding to the number of counterfactual outcome surfaces, and to adds a new Kronecker product corresponding to the outcome dimension, and (2) using the DKL structure to reduce the dimension of the GP on the covariate space, perhaps making the expanded covariance structure more tractable. I think focusing more of the paper on precisely describing the scaling challenges that this introduces and how DKL addresses them would be more appropriate. As of now, these technical contributions are difficult to pull out, where more than half of the paper is dedicated to restating known results.
>
> Slimming down the background material add adding more empirical exploration could be a useful way to reframe the paper.
>
> Some examples of unnecessary exposition:
>
> &nbsp;
>
> •	Figure 1(b) and 1(c) are unnecessary. The working assumption is that \vX satisfies the backdoor criterion, so discussing other cases is not relevant to this paper.
>
> **We have now removed Figure 1(b) and 1(c) and left Figure 1(a) only.**
>
> •	Theorem 2.2 is not necessary (it is also wrong, see below). Simply quoting the backdoor adjustment formula would be sufficient.
>
> **We have replaced the Theorem with a simple statement quoting the correct backdoor adjustment formula $ p(Y | do(A=a), X) = p(Y | A=a, X)$, as this was a typo, thank you.**
>
> •	Corollary 3.1 does not seem necessary. It is an asymptotic result, not finite sample, and the main point that it becomes hard to estimate an outcome surface when there are sparsely sampled input regions for a particular counterfactual does not require a formal statement. In fact, the target of estimation in this paper is not causal effects, but the intervention distributions themselves, so the primary content of this corollary (which relates the difficulty of estimating a contrast between surfaces to the difficulty of estimating each individual surface) is not relevant to the problem being solved. In terms of the challenges that result from many levels of A, simply stating that the sample size gets sharded across these actions, often very unevenly, seems to make this point effectively.
>
> **It is true that this work places particular emphasis on estimating the multivariate interventional distribution $ p(Y | do(A=a), X)$. However, the estimation of causal effects is a closely related task, for example in the form of CATE estimation, defined through moments of this distribution as $ \tau_{a, b} (\mathbf{x}_i) = \mathbb{E} [Y_i | do(A_i = a), \mathbf{X}_i = \mathbf{x}] - \mathbb{E} [Y_i | do(A_i = b), \mathbf{X}_i = \mathbf{x}]$.**
>
> **Corollary 3.1 is indeed an asymptotic result about the difficulty of estimating CATE with any nonparametric regression method, but it formally relates the optimal (minimax) rates of convergence to both the problem of having multiple actions and high-dimensional covariate space in a principled way.**
>
> **We have decided to keep Corollary 3.1 in and have instead rephrased part of Section 3.1 to better clarify its relevance. We are open for a follow-up discussion with the reviewer about this.**

---

> > ### Author Response · Authors · 2022-09-09
> > **Response to Reviewer 9sDy's comments (Part 2)**
> >
> > # Response to Reviewer 9sDy's comments (Part 2)
> >
> > &nbsp;
> >
> > ## Exploring bias-variance tradeoffs
> >
> > &nbsp;
> >
> > The authors note that the primary benefit of the multi-task GP framework is sharing information across outcome surfaces (either between counterfactuals or between outcome dimensions). However, this information sharing introduces bias. Right now the paper does not do a good job of noting the kind of risk that this bias can introduce, and the settings under which the bias would be beneficial. For example, in Figure 2, the red surface could have just as easily continued horizontally, in which case the multi-task GP estimate would be very far off.
> > One additional risk here is that the residual correlation between counterfactual outcomes---say, Y_a1, Y_a2---after conditioning on X is not identifiable, even with infinite data, but the proposed approach (and CounterGP) specify a correlation between them. If this specification introduces bias in the estimated intervened expectation surfaces, it would not fade away with larger sample size. This would also certainly affect the joint uncertainty bands associated with Y_a1 and Y_a2.
> >
> > It would be useful to specify some simulations where the multi-task GP helps and some where it hurts (at least in finite samples), and see how the benefit fades with larger sample size.
> >
> > **This is true. As already mentioned in the last paragraph of Section 3.1, taking a multitasking approach in causal learning implies assuming a priori that the underlying potential outcomes surfaces share similar patterns, such as in Fig. 2’s example (for example, assuming that CATE displays simple patterns of heterogeneity with respect to prognostic effects Hahn et al. (2020), Caron et al. (2022a)). However, in settings where this is not true, performance of multitask GP may deteriorate compared to a T-type of Learner (learning the two surfaces with two independent GP regression).**
> >
> > **To illustrate this point, we will add a simple one-confounder toy example, similar to the one in Figure 2, where instead “the red surface continuous straight” and the two surfaces do not share similar patterns.**
> >
> > &nbsp;
> >
> > ## More detailed empirical exploration
> >
> > &nbsp;
> >
> > The empirical exploration in this paper could be much stronger.
> >
> > Evaluation of Bayesian uncertainty from the model, and evaluations on a wider range of simulations that showcase different aspects and tradeoffs of model performance would be a good start.
> >
> > Also, specifying a problem where the multi-outcome mutli-task GP clearly helps would be useful.
> >
> > **We are working on a new experiment in Section 5.1 to demonstrate performance of (multioutput) CounterDKL on the task of Individual Causal Effects (ICE) estimation, in terms of coverage for uncertainty quantification.**
> >
> > **In addition, in the same new simulated example, we introduce a new parameter $\gamma$ that governs the degree of confounding, so that we can investigate models' performance (RMSE and uncertainty quantification), and sample efficiency of CounterDKL and MODKL, for increasing levels of confounding (i.e. for increasing values of $\gamma$).**
> >
> > Finally, demonstrating the method on a real dataset that has the multi-action-level multi-outcome structure would provide far stronger motivation. The authors claim that this is a common setting, so it should be doable to find such a dataset.
> >
> > **The work on CounterDKL in this paper has been inspired by an ongoing collaboration the authors are taking part in. The project is a cardiovascular study involving observational patient-level data, and features high-dimensional covariate space, multi-category treatment and two correlated outcomes (risk of myocardial infarction and risk of bleeding), where treatment decreases the risk of one but increases the risk of the other condition Unfortunately, owing to the nature of the real dataset (both practical and ethical considerations), that data cannot be utilized for this current paper.**
> >
> > **In response, we have designed the experiments in Section 5.1 to closely mimic this type of study, with similar categorical covariates and correlated outcomes.**
> >
> > **We have also added a mention to this in the introductory section.**

---

> > > ### Author Response · Authors · 2022-09-09
> > > **Response to Reviewer 9sDy's comments (Part 3)**
> > >
> > > # Response to Reviewer 9sDy's comments (Part 3)
> > >
> > > &nbsp;
> > >
> > > ## Misstatements in Section 2 on Causal Identification Conditions
> > >
> > > &nbsp;
> > >
> > > A DAG is not equivalent to an SCM. The DAG does not specify the F or p(u) part of the tuple. A better statement would be that an SCM implies a DAG.
> > >
> > > **Indeed. We have corrected this typo, thank you.**
> > >
> > > The statement of Theorem 2.2 is incorrect. The backdoor adjustment formula says that p(\vY | do(A=a), \vX) = p(\vY | A=a, \vX). From this P(\vY | do(A=a)) is recovered by E[p(\vY | A=a, X)], where this expectation is taken over the marginal distribution of X, p(X). Note that P(\vY | A=a) = E[p(\VY | A=a, X) | A=a], which is very different, even if the backdoor criterion is satisfied. The last equality in the proof is wrong (it would only hold if p(x | A=a) = p(x)).
> > >
> > > Theorem 2.2 is also unnecessary. The statement of the do calculus puts no restrictions on the dimensionality of the variables in the graph, so this statement for multivariate Y is already implied by well-established results.
> > >
> > > **We have amended this (corrected and replaced theorem with just a statement). See previous comment above (first section).**
> > >
> > > “Provided that covariates X (or a subset of them) satisfy the backdoor criterion”. Note the famous M-bias or M-structure example in which a subset of X satisfy the backdoor criterion, but adjusting for all of X does not. I.e., satisfaction of the backdoor criterion of a subset of variables does not imply that the superset satisfies the backdoor criterion.
> > >
> > > **We agree this statement as it is could cause confusion, given the case of collider bias. We corrected this by implying that the conditioning set X is made of direct common causes/parent of A and Y.**
> > >
> > > &nbsp;
> > >
> > > ## Other Comments
> > >
> > > &nbsp;
> > >
> > > The quantity Y_i|do(A_i=a_i) is usually written as a counterfactual Y_{a} or a potential outcome Y(a). do notation is reserved for distributions (hence its use as a conditioning variable).
> > >
> > > **True, we have amended this, thank you.**
> > >
> > > The LHS of the definition equation right before (4) does not really parse. Should we be conditioning on f_Y, or on m(.) and K(.,.)? If conditioning on f_Y, it seems like f_Y(x*_j) is just a constant. Meanwhile, eq (4) is written entirely in terms of the hyperparameters m and K.
> > >
> > > The indexing of the coregionalization matrix is extremely difficult to parse. In the end, since Q is fixed to be 1, it seems like it would make sense to just present the intrinsic coregionalization matrix here, and note that it can be generalized to the LMC.
> > >
> > > The LHS’s of the math environment on page 5 don’t seem to be consistent. In the second equation, should the arguments of k(.,.) be f_d(\vx_i) or should they just be \vx_i? It seems like the latter, given than the RHS is written in terms of \vx_i.
> > >
> > > **In response also to other reviewers’ comments, we have now substantially re-written the corresponding paragraphs of Section 3.1, in order to amend some confusing notation and to better clarify the separable kernel multitask paradigm.**
> > >
> > > ---
> > >
> > > ### Thank you!

---

### Decision · Action_Editors · 2022-11-07

**Recommendation:** Accept with minor revision

**Comment:**

Two out of three reviewers noted the updated version sufficiently addressed their comments.

The third review is more critical, especially on the issue of bias-variance tradeoffs: "...poses a number of modeling assumptions that introduce bias (in particular, bias that does not go away even with infinite samples) in order to make estimation more sample efficient. However, there is little to no discussion in the paper about how users of the method should reason about these design choices. Unlike standard ML settings, one cannot empirically calibrate bias-variance tradeoffs on causal estimation problems because ground truth individual causal effects are not observable. Thus, it is necessary to provide users with a picture of when modeling choices will succeed and, critically, when they will fail so that users can reason about the suitability of the method for their application. Neither the method development nor the experiments provide adequate resolution on this core issue."

My understanding is that this issue has been acknowledged by the authors, but perhaps some recommendations could be added based on empirical results to guide practitioners.

One reviewer raised concerns about the new contribution list. I agree with this and would suggest having only the contributions in the list, not the background/structure of the paper.

All reviewers note the incremental modelling changes compared to prior work. However, as the paper is technically solid/correct and could be of interest to the TMLR community, I suggest accepting it with minor revision.


**Audience:**

Yes, the audience at the intersection of GPs and causal inference could find this interesting. However, the novelty factor is not high, as noted by all reviewers.

**Claims And Evidence:**

Yes. In my view, the updated version and the answers have sufficiently addressed the reviewers' concerns.